# Action and Entropy in Heat Engines: An Action Revision of the Carnot Cycle

**DOI:** 10.3390/e23070860

**Published:** 2021-07-05

**Authors:** Ivan R. Kennedy, Migdat Hodzic

**Affiliations:** 1School of Life and Environmental Sciences, Sydney Institute of Agriculture, University of Sydney, Sydney, NSW 2006, Australia; 2Faculty of Information Technologies (FIT), University of Mostar, 88000 Mostar, Bosnia and Herzegovina; migdathodzic@gmail.com

**Keywords:** Carnot cycle, caloric, specific heat, entropy, Gibbs potential, vortical entropy, reversible cycle, working fluid, quantum field, relative action, heat engine

## Abstract

Despite the remarkable success of Carnot’s heat engine cycle in founding the discipline of thermodynamics two centuries ago, false viewpoints of his use of the caloric theory in the cycle linger, limiting his legacy. An action revision of the Carnot cycle can correct this, showing that the heat flow powering external mechanical work is compensated internally with configurational changes in the thermodynamic or Gibbs potential of the working fluid, differing in each stage of the cycle quantified by Carnot as caloric. Action (@) is a property of state having the same physical dimensions as angular momentum (*mrv* = *mr*^2^*ω*). However, this property is scalar rather than vectorial, including a dimensionless phase angle (@ = *mr*^2^*ωδφ*). We have recently confirmed with atmospheric gases that their entropy is a logarithmic function of the relative vibrational, rotational, and translational action ratios with Planck’s quantum of action *ħ*. The Carnot principle shows that the maximum rate of work (*puissance motrice*) possible from the reversible cycle is controlled by the difference in temperature of the hot source and the cold sink: the colder the better. This temperature difference between the source and the sink also controls the isothermal variations of the Gibbs potential of the working fluid, which Carnot identified as reversible temperature-dependent but unequal caloric exchanges. Importantly, the engine’s inertia ensures that heat from work performed adiabatically in the expansion phase is all restored to the working fluid during the adiabatic recompression, less the net work performed. This allows both the energy and the thermodynamic potential to return to the same values at the beginning of each cycle, which is a point strongly emphasized by Carnot. Our action revision equates Carnot’s *calorique,* or the non-sensible heat later described by Clausius as ‘work-heat’, exclusively to negative Gibbs energy (−*G*) or quantum field energy. This action field complements the sensible energy or vis-viva heat as molecular kinetic motion, and its recognition should have significance for designing more efficient heat engines or better understanding of the heat engine powering the Earth’s climates.

## 1. Introduction

“Partout où il existe une differénce de temperature, il peut y avoir production de puissance motrice.”

Sadi Carnot’s authoritative statement regarding the dependence of power in heat engines to a gradient in temperature founded modern thermodynamics. The fully reversible cycle for ideal gases as working fluid in a heat engine, proposed by Carnot in 1824 [1,2], led to a revolution in our understanding of heat, energy, entropy, and their relationships. This reversible cycle requires both efficient heat flows during isothermal processes and perfect insulation during adiabatic processes when no heat flows occur. For maximum efficiency, all heat input to the cycle is considered to be at the temperature of the source, while heat rejected is considered to be at the temperature of the sink. Carnot recognized that it is impossible for all the heat taken from the hot source to be converted to external work; part of the input heat must be dissipated at the colder sink to allow a cyclic process that restores the system to its original state. In modern terms, it is essential that the entropy of the working fluid, as a property of state, be restored to its original value once each cycle is completed. In practice, the working fluid of a heat engine does not cycle ideally between these two temperatures because of the limited speed of temperature equilibration between the heat source or sink and the working fluid. The ideal Carnot cycle is the most efficient possible because heat is transferred reversibly and isothermally only while the working fluid is at the same temperature as the source or the sink. Furthermore, Carnot proposed that the motive power of the engine is also a function of product of the volume of the working vapor and its pressure at the extremes of temperature. Note that his cycle actually estimates the maximum work possible thermodynamically rather than the motive power or rate of work.

However, he also claims that the rate of work performed in the cycle is not defined by differences in kinetic energy or pressure of the working fluid at the temperature extremes but by internal forces related to the varying content of caloric. These changes during isothermal and adiabatic expansions and compressions exactly compensate each other as required to complete the cycle. Yet the pressure and volume changes in the cycle are essential, providing the forces required to allow the conversion of heat to work isothermally.

## 2. Revisiting the Carnot Cycle

The reversible Carnot cycle [3,4] consists of the following four stages.

Stage 1=>2. An isothermal expansion at the temperature (*T*_source_) of a hot source (such as a coal fire in a steam engine) during which heat is transferred from the source to the working fluid and external work is done. This stage provides heat isothermally (we identify as Carnot’s caloric *a*) with equivalent work performed as a logarithmic function of the expansion in volume.

Stage 2=>3. An adiabatic expansion of the working fluid to the lower temperature of a heat sink (*T*_sink_) during which further external work is done at the expense of the heat content of the gases (consistent with Carnot’s caloric *b*′), but no heat enters or leaves the piston chamber.

Stage 3=>4. An isothermal recompression at the lower temperature of the heat sink during which work is done on the working fluid and heat is transferred from the working fluid to the cold sink (Carnot’s caloric *a*′).

Stage 4=>1. An adiabatic compression during which further work is done on the working fluid (Carnot’s caloric *b*) as it is reheated to the original temperature of the hot source.

Possibly never discussed subsequently, we state that Carnot in 1824 concluded that (*a* + *b*) must equal (*a*′ + *b*′) and that therefore, the maximum work possible is determined by (*a* − *a*′) or (*b*′ − *b*) [2]. In doing so, Carnot ruled out the equality of the heat transferred into the engine in stage 1=>2 from the hot source with that transferred out in stage 3=>4 to the colder sink, although Clapeyron wrongly quoted Carnot (see Dover edition [2]) as claiming equality. Carnot’s principle regarding the need for a temperature gradient is made clear in the following equation giving the maximum efficiency possible from the work cycle:[*R* ln(*V*_2_/*V*_1_)(*T*_source_ − *T*_sink_)]/[*R* ln(*V*_2_/*V*_1_)(*T*_source_)] = (*T*_source_−*T*_sink_)/(*T*_source_).(1)

Thomson [5,6] pointed out from Clapeyron’s account of Carnot’s conclusions that the efficiency of a heat engine was given by the following ratio:(*Q*_source_ − *Q*_sink_)/*Q*_source_ = (*T*_source_ − *T*_sink_)/(*T*_source_).(2)

Yet, however well-known these conclusions from Carnot’s cycle are, the full content of Carnot’s memoir on the cycle is rarely understood, lacking his appreciation of internal changes in the working fluid and the significance of the energy field he referred to as caloric on the morphology of the system. We present here a revision of the Carnot cycle informed by our developing theory of action mechanics.

## 3. Action and Entropy

The action resonance theory [4] includes a set of explanatory statements, which are based on the quantum of action as described by Planck and extended by Einstein. According to Planck, whenever a quantum of energy is absorbed, there is an equivalent increase in entropy. Einstein confirmed that the momentum of quanta is particulate, absorbed, and emitted whenever quanta are exchanged. The impulses of quanta interact within the conformation of molecular systems, sustaining its configuration in an energy field consisting of resonant quanta. Although the impulses from quanta are relatively small compared to those between heavier molecules, the greater speed of photons as quanta allows a greater frequency of impulses, compensating momentum exchange on variable radii. For ideal systems, the theory assumes no interactions between molecules of gas except in collisions; they move in independent trajectories defined by their radial separation. Kinetic or internal energy is a product of field interactions, which is in effect a result of torques exerted by the specific energy field that increases per molecule if work is done, increasing volume at constant temperature and compensating for the greater radius. This action revision has been developed from statistical mechanics, reinterpreting phase space (*dq*.*dr*) as differentials for action space (*mrv*). The partition functions of statistical mechanics have been reinterpreted as action ratios relative to Planck’s quantum of action (*mrv*/*ħ*), allowing exact calculations of thermodynamic properties such as entropy and free energy [7]. A feature is that at lower density, the field energy per molecule increases, reflecting the greater radial separation and molecular action. By comparison, at high molecular density, action per molecule is minimal as radius declines, corresponding to increased Gibbs and chemical potential.

A significant revision of the traditional approach to the Carnot cycle to include complementary variations in Gibbs potential in the cycle is proposed. Ignoring this complementarity between matter and energy that was understood by Carnot may have hampered the beneficial application of thermodynamics to the natural world. A focus on the kinetic heat and the vis-viva of the working fluid was adopted by Lord Kelvin [5], presumably to discard all traces of the caloric theory of heat as a permanent fluid. Carnot’s theory of heat engines was far more nuanced, rejecting *calorique* as permanent in heat value in the cycle but still having an essential quantitative role. Indeed, his insistence that the quantity of *calorique* (a) contributed from the heat source at higher temperature differed in magnitude from that absorbed by the colder heat sink (*a*′) foreshadows statistical mechanics and quantum theory, as this paper will show.

An alternative but consistent approach to the classical treatment of the Carnot cycle involves the calculation of entropy using the quantum property of action [4]. According to Kennedy et al. [7], the entropy (*S*) is naturally partitioned as translational (@_t_), rotational (@_r_), and vibrational forms (@_v_) forms of relative motion as action. This approach allows easier calculation of the entropy of ideal gases and the Gibbs energy, as shown in Section 5 and Section 6. To obtain the total entropic heat energy required to reversibly bring a gas from absolute zero (where the entropy is zero) to the current temperature, it is sufficient to multiply each of these partitioned entropies by the temperature *T* and then to take their sum (*ST*). This includes the heat required for all isothermal phase changes such as melting and vaporizing as well as for other forms of disaggregation. For simplicity of expression, each partition for entropy contains within the logarithm an exponential term that accounts for enthalpy (e.g., e^3/2^, e^5/2^) depending on the complexity and degrees of kinetic freedom of the molecule. A relative action ratio (e.g., @_t_/*ħ* = n_t_) is also included, indicating the mean quantum state or molecular configuration. The logarithm of this term accounts for latent heat in *ST* or negative Gibbs energy that varies with volume and temperature.

This approach seeks to simplify and generalize the complex statistical mechanical functions for entropy. In his book on elementary statistical mechanics of 1902, Gibbs [8] established the principle of conservation of extension in phase for systems in statistical equilibrium. He remarked that the differential product *dp.dq* for momentum and position has dimensions of energy multiplied by time. “Hence an extension-in-phase has the dimensions of the *n*th power of the product of energy and time. In other words, it has the dimensions of the *n*th power of action, as the term is used in the ‘principle of Least Action’”. He also defined the coefficient of probability or entropy as “the reciprocal of the extension-in-phase, that is the reciprocal of the *n*th power of the product of time and action”. Schrödinger [9] drew attention to the canonical partition function (*Z*) or sum over states (Σe^−*Ej*/*kT*^), indicating the total occupation number of different energy states as a fraction of the total number of possible molecular systems N. Then, *k*ln*Z* is equal to (*S*–*E*/*T*), and it can be shown that *Z* equals (n_t_^3^e)*^N^*, where n_t_ is the relative molecular translational action (@/*ħ* = *mrv* = n_t_) for each of N molecules of a monatomic gas. Note that quantum numbers n_t_ given are mean values that would be integral for particular molecules. Furthermore, the microcanonical or molecular partition function (*z*) is justified, using statistical mechanics, as equal to Nn_t_^3^ for N molecular systems, in an addendum attached to this paper. This Appendix A also considers the status of vibrational action comparing N_2_ to CO_2_ and how its excitation can be considered as the translational action of activated molecules.

In his account of statistical mechanics, Hill [10] (Equation 8.37) has expressed the average entropy per molecule of a diatomic gas such as N_2_, including the sensible heat or enthalpy (*H*) for constant pressure systems, as follows:(3)$$S/N=k\{\ln[(2\pi mkT/h^{2}{)}^{3/2}Ve^{5/2}/N]+\ln[\left(8\pi^{2}kTI_{r}/h^{2}\right)e/\sigma ]\;+[(h\nu /kT)/\left(e^{h\nu /kT}-1\right)-\ln(1-e^{h\nu /kT})]\}$$

Monatomic ideal gases such as helium or argon lack the second and third rotational and vibrational terms in Equation (3). This partitioned equation can be simplified using action mechanics [7], ignoring vibrational entropy that need not be considered here, given its relatively small magnitude for N_2_, even at the most elevated temperatures. However, its rotational entropy is highly significant.
*S/N = s = k* ln[e^7/2^{(3*kTI*_t_/*ħ*^2^)^3/2^/z_t_}(2*kTI*_r_/*ħ*^2^σ)](4)
(5)$$S/N=s=k\ln[e^{7/2}{\left({@}_{t}/\hbar \right)}^{3}\left({@}_{r}/\hbar {)}^{2}\right]$$

This more holistic expression is composed of thermal (3.5*k*) and statistical or configurational elements for translational action (@_t_ = *mrv = mr^2^ω* = *Iω*) and rotational action (@_r_ = *I*ω). The symbol @_t_ represents the relative translational action, which is a functional property of molecular momentum and radial separation equal to [(3*kTI*_t_)^1/2^/z_t_^1/3^] [7]; *I*_t_ is a translational moment of inertia calculated for a cubic distribution of molecules, and z_t_ (z_t_ = 10.2297) is a factor avoiding double counting of molecules and correcting for the ratio of their mean speed and their root-mean-square velocity. The moment of inertia for translational motion (*I*_t_) is equal to *mr*^2^—the molecular mass multiplied by the square of the mean radial separation of similar molecules. The one-dimensional translational action (@_t_) varies at each of the four stages of the Carnot cycle as pressure and temperature vary. The rotational action of two-dimensional molecules such as N_2_ is obtained as (2*kTI*_r_)^1/2^, where *I*_r_ is the moment of inertia of unvarying radius as in chemically bonded structures. Then, entropy action partitions are given in (6)–(9).
*S*_t_ = *R*ln[e^5/2^(@_t_/*ħ*)^3^] = *R*ln[e^5/2^(n_t_)^3^] (translation)(6)
*S*_r_ = *R*ln[e(@_r_/*ħ*)^2^] = *R*ln[e(j_r_)^2^] (rotation-diatomic or linear molecule)(7)
*S*_r_ = *R*ln[π^1/2^e^3/2^(@_A_@ _B_@_C_/*ħ*^3^)] (rotation-polyatomic molecule)(8)
Σ*S*_vi_ = Σ[*R*x/(e^x^ − 1) − *R*ln(1 − e^−x^)], where x = *hcν*_i_/*kT* (vibration of each bond)(9)

Thus, the total entropic energy or heat required at *T* is governed by the sum below (10).
*S*_Total_*T* = [Σ(*S*_t_ + *S*_r_ + *S*_vi_)]*T*(10)

Using this action approach, excellent agreement with standard third-law experimental measurements of entropy of atmospheric gases is obtained as the integral of heat added reversibly to reach the standard temperature of 298.15 K. Values correct to four significant figures at 1 atmosphere pressure for all gases in air have been estimated [7] and at their actual pressures. This accuracy is adequate for calculating equilibrium constants for gaseous chemical reactions even at elevated temperatures such as the extent of dissociation of hydrogen molecules on the Sun into hydrogen atoms (Kennedy, unpublished). For a Carnot cycle with a monatomic gas, only translational action need be considered to determine the entropy. In this paper, a heat engine using gaseous argon as the working substance is examined. However, the analysis is easily extended to polyatomic molecules [7], also shown here for dinitrogen, requiring consideration of rotational action as well as translational action to estimate entropy.

A translational symmetry constant z_t_ in Equation (4) corrects the magnitude of the translational action to match precisely the field energy required to sustain it; for the translation of ideal gases at 1 atmosphere pressure, this energy-sparing factor z_t_ is of constant magnitude of 10.2297 for all species of molecules [7,11]. The correction has now been logically interpreted [12] as involving inverted sub-factors of (1/2)^3^ to prevent double counting of molecular partners and (1/1.0854)^3^ to correct the root-mean-square velocity (from *kT* = *mv*^2^/3) to the mean molecular velocity, which is required to calculate the translational action (*mvr* = *mr*^2^*ω*). It is noteworthy that this means establishing molecular entropy in a reversible process as defined by Clausius [13], which requires that the entropy per molecule is dependent on gas density, increasing logarithmically as the mean volume (*a*^3^) occupied by each molecule increases. Such a relationship with density for heat content was also proposed by Carnot.

## 4. Gibbs Energy

Based on the theory above, it is possible to calculate the thermodynamic properties of the Carnot engine with the following empirical equations, using a monatomic working substance.
*S_n_T_n_* = *RT_n_*ln[e^5/2^(@_t_/*ħ*)*^3^*] = 2.5*RT_n_* + *RT_n_*ln[(@_t_/*ħ*)^3^] = *H_n_* + *RT_n_*ln[(@_t_/*ħ*)^3^](11)

Here, the enthalpy (*H_n_*) is equal to the molar internal energy (*E*_n_), plus the pressure-volume function (*RT_n_*) for atmospheric work, which is obligatory for a system open to air, although not in the Carnot cycle where internal pressure varies with volume, which is reversibly equal to external pressure. Negative signs are given for the two so-called free energies, which are actually inversed potential energies—the Helmholtz energy (*A*_n_) used in constant volume systems and the Gibbs energy (*G*_n_ or *g*_n_ per molecule) used with systems open to the atmosphere requiring pressure–volume work also affecting heat content. These calculations shown in (12) and (13) give exact values, not differences.
−*A*_n_ = *RT_n_*ln[e(@_t_/*ħ*)^3^] = *RT_n_*ln[e(n_t_)^3^](12)
−*G*_n_ = *RT_n_*ln[(@_t_/*ħ*)^3^] = *RT_n_*ln[(n_t_)^3^](13)

The mean values of relative action (@/*ħ*) can be estimated as mean quantum numbers n_t_ as shown in (12) and (13), which are related to the molecular Gibbs energy (*g*_t_) as an indicator of the field energy at temperature *T_n_*.
−*G*_n_/N = *kT_n_*ln(n_t_)^3^ = −*g*_t_(14)

Thus, we can express these two functions that indicate chemical potential [4] as positive in the inverse of their negative values, given the logarithmic relationship in (15).
*G*_n_ = *RT_n_*ln[(*ħ*/@_t_)^3^](15)
*A*_n_ = *RT_n_*ln[(*ħ*/@_t_)^3^/e] = *RT_n_*ln[(*ħ*/@_t_)^3^] − *RT_n_*(16)

Obviously, these free energies or functions for ideal gases have negative values, decreasing even further as the heat absorbed to support external work by the ensemble of molecules in cycle stage 1 increases entropy. From these expressions, we can also write the following thermodynamic relationships.
*G*_n_ = *A*_n_ + *RT_n_*(17)

For a monatomic gas where *C*_v_ is 1.5*R*, separating the Gibbs energy from the enthalpy, we have (18) at constant pressure.
*S_n_T_n_* = *RT_n_*ln[(@_t_/*ħ*)^3^] + 2.5*RT_n_* = −*G*_n_ + *H*_n_(18)

Exchanging the terms, we obtain more familiar equations *G*_n_ = *H*_n_ − *S_n_T_n_* or, to indicate spontaneous change, we have:Δ*G*_n_ = Δ*H*_n_ − Δ*S_n_T_n_*.(19)

To include diatomic gases such as nitrogen and oxygen, we need to include rotational action [7]:*S_n_T_n_* = *RT_n_*ln[e^7/2^(@_t_/*ħ*)*^3^*(@_r_/*ħ*)*^2^*] = *RT_n_*ln[(n_t_)^3^(j_r_)^2^] + 3.5*RT_n_* = −*G*_n_ + *H*_n_.(20)

Here, the relative rotational action @_r_ for a diatomic molecule is taken as equal to (2*kTI*_r_/σ)^1/2^, where *I*_r_ is the molecular moment of inertia and σ is a symmetry factor preventing excess counting of indistinguishable conformations; this factor has the value of 2 for diatomic molecules such as N_2_, where each end of the molecule presents the same but reaches 12 in the case of methane (CH_4_, σ = 4!/2!). The symmetry indicated by σ is considered as a statistical factor adjusting for the likelihood of an encounter by a quantum of energy with an indistinguishable species of molecule that is proportional to its symmetry. Since there is no way to distinguish one end of an N_2_ molecule from the other, except isotopically, the concentration of such symmetric molecules is effectively doubled by comparison with NO, and the distance and elapsed time between encounters shortened. Any two systems having the same difference between enthalpy and entropic energy (*H*-*ST*) will have the same Gibbs free energy (*G*) and will be at equilibrium if opposed to each other. We should be aware that mechanistically, the negative-entropy energy term (−*S_n_T_n_*) contains both the other terms, so Equation (19) can be seen as a tautology. Some of the confusion regarding the nature of molecular entropy results from a lack of awareness of this fact. Equations (18) and (20) are more informative—they express the heat content of a polyatomic gas as the sum of the latent or potential heat, a logarithmic function of pressure or volume and temperature, plus the sensible heat or enthalpy, which is a function of the temperature alone.

The negative Gibbs energy term (−*G*_n_) expresses the non-sensible heat content, whereas the enthalpy term expresses the sensible or kinetic heat. The influence of both these functions was clearly identified by Carnot [1,2]. The Gibbs and Helmholtz functions are greatest when the internal potential energy is least, conversely to the entropy. Thus, higher quantum states, achieved as more quanta are absorbed by the field, correspond to increased entropy and decreased free energy, as stated by Planck [14]. By contrast, molecules in their ground states at the lowest temperatures have minimum entropy. Paradoxically, the Gibbs energy or function is only a potential to acquire field energy and is greatest when the latter is least. Often referred to as free energy—perplexing generations of students—this is actually true in the sense of being a measure of a molecule’s inaction and relative freedom from sustaining field energy. Its alternative name as recommended by the IUPAC of the Gibbs function is a neutral description, but we prefer Gibbs potential as even more descriptive. The following Section 5 and Section 6 summarize the key results of this paper.

## 5. An Action-Based Calculation for the Carnot Cycle

Using action mechanics, we can now describe the following action features corresponding to the four stages of the cycle for both argon and nitrogen gases as working fluids.

### 5.1. Isothermal Stage 1=>2

The traditional expression [3,4] of the isothermal stage 1 of the Carnot cycle (per mole) is given as Equation (21).
*Q*_source_ (1) = −*W*_rev_ (1) = *RT*_source_ ln*(V*_2_*/V*_1_*)*; And Δ*S* (1) = *Q*_source_ (1)/*T*_source_(21)

This is considered as the heat required to perform a reversible expansion against a variable external pressure. The work done in an expansion is a logarithmic function of the volume ratio as shown in (21). As discussed above, in an isothermal system, the translational action per molecule (@_t_ = *mr*^2^*ω*) varies proportional to *k*ln(*r*^3^) as the volume increases. Then, the external work per molecule is equal to the internal configurational or quantum work—the mean decrease in translational Gibbs energy per molecule (-δ*g*_t_). The convention that internal work implies a change in Gibbs potential is applied. We have:−*w* = *kT*_source_ln[(*r*_2_)^3^*/*(*r*_1_)^3^] = *kT*_source_ln[@_t2_/@_t1_]^3^ = *kT*_source_ln[n_2_/n_1_]^3^ = −δ*g*_t_.(22)

This is the decrease in translational chemical potential during the expansion process. In the isothermal reversible absorption of heat from the high-temperature source, the total heat work per molecule (δ*sT*) increases by the amount of heat added. That the work varied with the logarithm of the volume was clearly identified as operative in the cycle by Carnot [2]. Since molecular rotation is a function only of temperature and not pressure, there is no need to assign rotational heat energy as a variable in this process. Isothermally, pressure varies with volume (i.e., *pa*^3^ = *kT* = *mv*^2^/3 where *V* = N*a*^3^ and *v* is the root-mean-square velocity), so that in an isothermal expansion, the product of pressure and specific volume (*pv*, *pa*^3^) remains constant with the increasing volume per molecule, while the pressure decreases from its maximum. Performing work reversibly requires that the external pressure or mechanical resistance should always be equal to the internal pressure.

Pressure at constant temperature is a function of kinetic energy per unit volume. Action resonance states [4] that sustain molecular kinetic energy require sufficient molecular torque exerted by the intensity of field energy appropriate for the temperature required. This requirement is a logarithmic function of volume. This stage involves no change in internal energy (c_v_δ*T*), as the temperature remains constant and the kinetic energy is a function of temperature independent of volume for an ideal gas. The intensity of field quanta required to maintain a constant temperature while external work is being done is a logarithmic function of the increase in volume, as Carnot stated prominently in his memoir. The amount of heat *Q*_source_ is acquired by the working fluid, and the resultant increase in entropy is *Q*_source_/*T*_source_ (see also Table 1 and Table 2 below). Carnot designated *Q*_source_ equal to an amount of caloric *a* [2]; the external work performed in his discussion is shown in Equations (21) and (22).

Thermal radiation is not normally considered in the modern Carnot cycle. This isothermal increase in quantum state is paid for by the heat absorbed from the source *Q*_f_, which can be equated to the internal change in action (@) shown in 3*kT*_f_ln(@_2_/@_1_). This heat has been absorbed to sustain the working fluid in its new higher quantum state. For a reversible system, more field energy is needed to sustain molecules with greater spatial separation if external work is done.

### 5.2. Adiabatic Stage 2=>3

The traditional Carnot cycle version for this stage is given simply as a function of the difference in temperature by the heat capacity. For argon, this is 1.5*RT* per mole or 1.5*kT* per molecule. The external work done is considered as equal to the fall in kinetic energy as the system expands, reducing the temperature. This results in:−*W* (2) = *C*_v_(*T*_source_ − *T*_sink_).(23)

Closer examination reveals that while external work is done according to Equation (23), there is also an internal compensation of entropy increase from increase in volume matching the decrease in temperature, giving an overall change in entropy of zero. We have:*R* ln*(V*_3_*/V*_2_*)* + *C*_v_ln*(T*_sink_*/T*_source_*)* = Δ*S = 0*.(24)

In this adiabatic stage, when no heat enters or leaves the system, the external work performed requires kinetic heat from within the system, thus reducing the temperature to that of the sink. However, it is clear that the terms involving ratios of volume and temperature compensate for each other exactly. For argon, the increased volume means that the molecules are more separated, increasing the action state, while the decrease in temperature decreases the action state to the same extent. Overall, the action and thus the entropy remain the same, so there is no change in translational quantum number (@_t_/*ħ* = n_t_), despite the fall in temperature. Thus:δ@ = (*mr*_3_^2^*ω*_3_) − (*mr*_2_^2^*ω*_2_) = 0.(25)

Even though there is no change in action or entropy, the change in temperature causes a change in Gibbs energy as the internal work shows in Equations (26) and (27).
−*w*_internal_ = *kT*_sink_ln[(*mr*_3_^2^*ω*_3_)/*ħ*)]^3^ − *kT*_source_ln[(*mr*_2_^2^*ω*_2_)/*ħ*)]^3^(26)
= *k*(*T*_sink_ − *T*_source_)ln[(n_t_)]^3^ = −δ*g*_t_(27)

For an ideal diatomic gas such as N_2_, to include rotation as well as translation in the reversible adiabatic process, we will have the quantum number product shown in (28).
[(n_t_)^3^(j_r_)^2^]_sink_ = [(n_t_)^3^(j_r_)^2^]_source_(28)

This reflects the fact that more translational and fewer rotational quanta are required for a gas at the cooler temperature of the sink than at the temperature of the hotter source, although the associated quanta are smaller for both at the lower temperature. If we restrict the external work to the change in Gibbs energy for N_2_, we will have the following equation:−*w*_total_ = *k*(*T*_sink_ − *T*_source_)ln[(n_t_)^3^(j_r_)^2^] = −(δ*g*_t_ + δ*g*_r_).(29)

External work is possible in both stages 1 and 2, and this was also proposed by Carnot to occur reversibly to give maximum efficiency. For reversibility, the external back pressure by the object having work performed on it would need to be continuously equal to that given by the pressure in the working fluid. In the reverse stages 3 and 4, the heat engine or an external weight is performing the same pressure–volume work on the working fluid. Carnot refers to the caloric removed from the working fluid as work in stage 2 as equal to *b*′, as shown in Equations (27) and (29) as the Gibbs energy changes.

### 5.3. Isothermal Stage 3=>4

Having completed the expanding stages 1 and 2, the inertial effect of the heat engine continues for two compressive stages where work is performed on the gas by the engine. Heat *Q*(3) is extracted from the gas by the sink at its lower temperature. The lower the *T*_sink_, the lower the heat that is extracted during the compression and the greater the external work that is possible in each cycle. Hence:*Q_rev_* (3) = −*W_rev_* (3) = *RT*_sink_ ln(*V*_4_*/V*_3_); And Δ*S* (3) = *Q_rev_* (3)/*T*_sink_.(30)

Similar to stage 1=>2, stage 3=>4 is isothermal, continuing internal mechanical work performed in stage 2 shown by the increased Gibbs energy, as a result of extracting caloric or latent heat to the cold sink at air temperature (*T*_r_), although without changes in internal energy. We have:*−w* = *kT*_sink_ln[(*mr*^2^_4_ *ω*_4_)*/*(*mr*^2^_3_*ω*_3_)]^3^ = *kT*_sink_ln[@_t4_/@_t3_]^3^ = −δ*g*_t_ = δ*sT*_sink_.(31)

The Gibbs potential of the working fluid continues to increase while the total entropic energy (*sT*) declines even further. This corresponds to a decrease in the action. For this stage, we have *Q*_r_ = 3*kT*_r_ln(@_4_/@_3_), which has a negative value as the action declines to less than half. Note that @_4_/@_3_) is equal to @_1_/@_2_, so the changes in entropy for stages 1 to 2 and for stages 3 to 4 are equal but opposite. Given that the temperature is much greater for the first of these changes, the same change in relative action states costs much less in energy in stage 3 to 4. Carnot designated *Q*_r_ or 3*kT*_r_ln(@_4_/@_3_) in his discussion as *a*′, the caloric removed by the refrigerator body B, in stage 3=>4.

### 5.4. Adiabatic Stage 4=>1

The final isentropic, adiabatic, stage results in compressive work restoring internal energy lost in stage 2, with molecular c_v_ for monatomic gases like argon of 1.5*k*
−*w*_external_ = c_v_*T*_source_ − c_v_*T*_sink_(32)

Equation (32) exactly balances the decrease in kinetic energy per mole in Equation (23). On this basis, including the work done during changes in pressure, Kelvin’s argument that there was no need to invoke the idea of caloric in the heat engine cycle was made. However, we show there is also a loss of Gibbs potential as gas molecules are reheated by compression. In contrast to the traditional treatment of the adiabatic stages of the Carnot cycle where external work (*w*) is assigned to internal energy changes (Δ*E*) only, internal work processes as Gibbs energy changes in the working fluid also occur in stage 4=>1 as a result of the increase in temperature. By contrast to stage 2=>3, where Gibbs energy or potential is gained as temperature falls, Gibbs energy is lost in stage 4=>1, although the loss is less than the gain in stage 2=>3. Overall, this ensures that the entire cycle results in a zero balance in Gibbs energy for the working fluid. We have:*w*_internal_ = *kT*_sink_ln[(*mr*^2^_1_*ω*_1_)/*ħ*)]^3^ − *kT*_source_ln[(*mr*^2^_4_*ω*_4_)/*ħ*)]^3^= [*T*_source_ − *T*_sink_]*k*ln(n_t_)^3^ = −δ*g*_t_(33)

Given that the translational action remains constant in stage 4, the decrease in Gibbs energy is purely a function of the increase in temperature.

For diatomic molecules such as N_2_, both translational and rotational action will change as temperature and pressure increase. We also have the increase in internal energy with molecular c_v_ of 2.5*k*, which accompanies the decrease in Gibbs energy and the increase in entropic energy:*w*_internal_ = c_v_*T*_source_ − c_v_*T*_sink_.(34)

However, just as in stage 2=>3, the changes in quantum states exactly offset each other so that the Gibbs potential is proportional to the difference in temperature. Although the Gibbs energy per Kelvin [−*k*ln[(n_t_)^3^(j_r_)^2^] remains constant, the Gibbs energy declines. Hence:*w*_internal_ = *k*ln [1/(n_t_)^3^(j_r_)^2^](*T*_source_ − *T*_sink_) = (δ*g*_t_ + δ*g*_r_).(35)

Note that inversion of the quantum numbers makes internal work positive. This is the result whether work is considered in terms of mean Gibbs (*g*) or as total entropic energy per molecule (*sT*). Pressure or volume and temperature changes occur in these stages that exactly offset one another so they are isentropic, as shown in the tables following. However, despite the fact that no thermal heat enters or leaves the working substance in these adiabatic stages, inertial changes in the heat engine cause large conversions of latent and sensible heat into work and the reversal in stages 2 and 4 of work into heat in each cycle—a response to the change in temperature (*s*δ*T*). This reflects reversible work against gravity or as kinetic energy of shaft work linked to energy changes in the working fluid. So, gravitational energy as elevated weight or inertial potential energy of a flywheel stored reversibly outside the heat engine must also be an essential part of the Carnot cycle.

Carnot refers to the amount of caloric restored in stage 4 of adiabatic compression as *b*. As shown in Table 1 below, there is an increase in the Gibbs energy during the adiabatic expansion in stage 2=>3 that is much larger than the decrease that occurred in stage 1=>2 when heat was spontaneously transferred from the source as work was performed. However, the decrease in Gibbs energy in stage 4=>1 (*b*) is less than the increase in stage 2=>3 (*b*′).

An outline of the program employed to calculate thermodynamic outputs is shown in Figure 1.

## 6. Results: Carnot Cycle Calculations for Argon and Nitrogen

In Table 1 and Table 2, the results are shown from running the four stages of one complete Carnot cycle with specified heat inputs per molecule, using the action mechanics described in Section 3 to calculate entropy, Gibbs potential, and other properties of state at each stage. This experiment in Table 1 involves monatomic argon as a working fluid, operating with three degrees of translational freedom cycling between temperatures of 640 K (*T*_f_) and 288 K (*T*_r_)—the Earth’s average surface temperature as the coldest refrigerator or sink usually available.

It is important to understand that the tabulations refer primarily to the thermodynamic or quantum states of the molecules in the working fluid—not to the external work being done, although the maximum external work possible is shown in Table 1 as well. Table 2 provides similar results for nitrogen (N_2_), which is a molecule with five degrees of freedom by including rotation and operating between 640 and 288 K, which is the average temperature. The tables also contain calculated data related to thermal energy content, pressure, volume, action, Gibbs energy, entropy, and internal energy. Using action ratios, it is also possible to express the negative Gibbs energies or their equivalents in field energy as quantum numbers n_t_ and j_r_.

Some of the data in Table 1 and Table 2 are illustrated graphically in Figure 2 and Figure 3.

A monatomic gas such as argon has no vibrational degree of freedom capable of absorption or emission of these quanta. In addition, Table 1 shows that each molecule in its specific volume *a*^3^ is accompanied by a mean translational quantum number of 80.042. At 640 K after isothermal expansion, these quantum states have increased in density to a number of 111.708 per molecule. According to Clausius’ definition of entropy for reversible heat exchange, δ*sT* is equal to *Q*_source_/*T*_source_, but this is also equal to 3*kT*ln(@_2_/@_1_), which is effectively a change in quantum state n_t_ of about 80 to 112, as shown in Table 1.

As Carnot proposed, the work possible is shown in Table 1 and Table 2 to be independent of the working fluid, although the extent of adiabatic expansion and compression where no net work is performed reflects the different heat capacities (*C*_v_) of argon and nitrogen, as shown in line 4a and 4, respectively. The heat absorbed to reach these thermodynamic states as internal work is now field energy sustaining these molecular temperatures and pressures. In action mechanics [4], heat is more than molecular motion but includes the field energy sustaining the molecular motion. One is not possible without the other. Given that the mean kinetic energy (*I_t_ω*^2^/2) of each species of molecule is the same at a particular temperature, the mean pressure exerted by each molecular species is inversely proportional to its specific volume (*a*^3^). Thus, *pa*^3^ = *kT* is an average statistical property of each ensemble of molecules. For a volume *V, pV* = *NkT*, with *V* equal to N*a*^3^ with N, which is the average number of molecules per unit volume. Where N is a mole of molecules, N*k* is equal to the gas constant *R*. For more realistic consistency with physical models, it is convenient to make thermodynamic calculations as mean values per molecule. Then, values per mole are easily calculated multiplying by Avogadro’s number (6.022 × 10^23^). Note that the terms (*3kTI*_t_)^1/2^ and (2*kTI*_r_)^1/2^ used to calculation translational and rotational action include temperature (*T*) and radius (*r*_t_); the latter can act as a surrogate for the mean specific volume of each molecule. The ideal gas equation *p* = *kT*/*a*^3^ or N*kT* where N indicates number density can be used to substitute for variations in temperature, number density, or volume and pressure.

## 7. Discussion and Key Points

### 7.1. Sadi Carnot’s Legacy

Carnot [1] stated firmly on page 29 of his book regarding isothermal expansion, “When a gas increases in volume in geometrical progression, its *chaleur specifique* increases in arithmetical progression”. His statement regarding the logarithmic increase in specific heat with volume at constant temperature is consistent with the decreasing Gibbs energy or increasing entropic energy (δ*sT*) shown for stage 1 in Table 1 and Table 2, while internal energy (*c*_v_*T*) remains constant. The heat absorbed isothermally from the hot source is regarded as consumed in the field energy sustaining the molecular orbits and maintaining the kinetic energy of the molecules as constant, allowing external work to be done via their pressure. This configurational entropic energy was also defined by Clausius in 1875 [10] as work heat or the *ergal*. There was no need for the editor Mendoza (see his foreword in Dover edition of Carnot’s book [2]) to have rejected the significance of Carnot’s conclusion on page 29 of his memoir that the *chaleur specifique* (specific heat) varied with the logarithm of the volume. Indeed, the editor of the Dover edition, Mendoza, claimed in 1960 that Carnot was mistaken, having been misled by faulty data produced by Delaroche and Bérard to calculate the effect of pressure on the specific heat of a gas. In fact, Mendoza’s criticism of their data was in error, as discussed next.

Moreover, Carnot’s hypothetical table on page 33 of pressure varying from 1/1024 to 1024 atmospheres labeling variations in specific heat with pressure is correct in principle —if Carnot’s *chaleur specifique* is interpreted as variation in the heat required for entropic energy (δ*sT*) while work is performed—varying logarithmically with volume, decreasing Gibbs energy as a result of the increase in volume at constant temperature (stage 1=>2). When Carnot visualized the thermodynamic operation of the *motrice de feu*, he actually challenged the theory casting *calorique* as a diffusable fluid form of heat that could neither be created nor destroyed. He did this by proposing that *calorique* as heat could be temporarily exchanged with motive power in a reversible cycle. It is clear from what Carnot wrote that he did consider sensible heat or *chaleur* disappeared as internal work was performed in an adiabatic expansion.

At no point in his reflection does Carnot claim that the same quantity of heat as that introduced from the hot source is fully re-absorbed as caloric after the adiabatic fall (*chute de calorique*) to the temperature of the cold sink, which was an error introduced by Clapeyron [15]. On the contrary, on page 28 in the Mendoza edition [2] (page 29 in the 1878 Gauthier-Villars reproduction of the original Bachelier 1824 edition), he clearly infers to the quantity of caloric *a*, which is “necessary to maintain the temperature of the fluid constant during dilatation”, and that transferred from the hot source in stage 1 is not equal to the caloric *a*′ that the gas abandons later as a result of its reduction of volume at lower temperature in stage 3, so *a* − *a*′ must have a positive value.

### 7.2. Caloric as Negative Gibbs Potential

To examine his analysis here in more detail, Carnot [2] defines (p. 28, 30) the high-temperature phase of the cycle involving body A as consisting of two portions of caloric—that needed to maintain the temperature A in dilatation (*a*) and that needed to restore the temperature of the fluid from that of body B to that of body A (*b*). “The total caloric furnished by the body A will be expressed by *a* + *b*. The caloric transmitted by the fluid to the body B may also be divided in two parts: one, *b*′, due to the cooling of the gas by the body B; the other, *a*′, which the gas abandons as a result of its (isothermal) reduction in volume. The sum of these two quantities is *a*′ + *b*′; it should be equal to *a* + *b*, for, after a complete cycle of the operations, the gas is brought back exactly to its primitive state. It has been obliged to give up all the caloric which has been furnished to it”. So, we have:*a* + *b* = *a*′ + *b*′(36)
or rather
*a* − *a*′ = *b*′ − *b*.(37)

At this point, Carnot leaves unsaid that Equation (37) gives the maximum work possible in a perfect heat engine, the difference between two similar logarithmic functions of changes in volume but differing only by temperature. However, he clearly identifies [1] this link on page 22 with his precise statement: “La puissance motrice résultat …sera évidement la différence entre celle qui est produite par l’expansion du gaz, tandis qu’il se trouve á la temperature du corps A, et celle qui est consommeé pour comprimer ce gaz, tandis qu’il se trouve á la temperature du corps B”. Thus, he claims that the isothermal changes of volume at the different temperatures fully explains the potential to do motive work.

These terms can readily be identified in the results for the cycle stages 1–4 in Table 1 and Table 2, corresponding to the negative magnitudes of the Gibbs potential changes (or the Gibbs entropic energy values) in the columns 1=>2 (*a*), 2=>3 (−*b*′), 3=>4 (−*a*′), and 4=>1 (*b*). A typing error in his memoir on line 10 page 31 possibly first inserted in the 1878 edition of *b*′ for *b*, repeated in the Mendoza translation [2], may have been confusing. Note that modern convention requires that the signs of *a*′ and *b*′ be made negative given that the Gibbs energy is increased by the extraction of heat during the adiabatic expansion and the isothermal compression by body B. From this analysis, we can now identify changes in Gibbs potential with Carnot’s changes in caloric, with no need to include changes in energy (δ*e*) except in the stages where there is a change in temperature.

Carnot clearly understood that his term caloric related to the physical state of the working fluid with a meaning very similar to negative Gibbs potential or configurational entropic energy as calculated in action state theory. Whenever Carnot claims an increase in caloric of the working fluid, we can recognize absorption of radiant heat as increasing entropy and action and internal work of raising quantum states. Clausius refers to this reversible work as having consumed heat “nowhere present, it is consumed in the changes doing work”. Perhaps Clausius initially recognized Carnot’s perception as foreshadowing his *ergal*, being the internal work done on the engine’s working fluid. However, Clausius then seems to disregard the internal work implied for molecules further apart in the working fluid and decides to directly assign the heat required to external work.

Fortunately, despite Kelvin’s apparent misquotation from Clapeyron of Carnot’s theory implying that “all the heat from body A … during expansion has flowed into body B during compression” as work was done, in 1850, Clausius [13] kindly corrected this false viewpoint. The German inventor of the thermodynamic concept of entropy recognized the value of Carnot’s principle of motive work depending on heat being transported from a hot source to a colder sink, using a working fluid that was unchanged in its state. To emphasize that heat and work were interchangeable, Clausius later [15] abandoned the concept of latent heat, substituting ‘work-heat’ for increased entropy to account for the heat needed to overcome both the cohesion of molecules and expansion against an external pressure. Since these heat-absorbing processes are both reversible, heat disappears as work is performed, but work can reappear later as heat in a reversible Carnot cycle. Perhaps Clausius went too far in seeking to dispense with caloric when converting heat to work, many years before Planck and Einstein developed the theory of quanta.

It is surprising how the false statement from Clapeyron has persisted with highly skilled authors still continuing to claim that Carnot assumed all the heat given up by body A was transferred to body B [16,17]. Although Aumand et al. [17] creditably provide a correct account based on various sources, they still propagate the error that Carnot equated *calorique* to the property we now call entropy, which is one also fell into by the book editor Mendoza [2]. As discussed above, Carnot clearly meant energy, requiring the product (*s* × *T*) of entropy and temperature.

To put Carnot’s clear viewpoint to the contrary beyond doubt, his footnote in full on page 19 of the Mendoza edition of his book (page 20 of the 1878 Bachelier edition [1]), misquoted by Kelvin and Clausius, actually states “We tacitly assume in our demonstration, that when a body has experienced any changes, and when after a certain number of transformations, it returns to precisely its original state, that is to that state considered in relation to density, to temperature, to mode of aggregation… I say that this body is found to contain the same quantity of heat (*chaleur*) that it contained at first, or else that the quantities of heat absorbed or set free in these different transformations are exactly compensated. This fact has never been called into question. It was first admitted without reflection, and verified afterwards in many cases by experiments with the calorimeter. To deny it would be to overthrow the whole theory of heat to which it serves as a basis”. Thus, Carnot never inferred the equality of the heat provided from the hot source to that retrieved by the cold sink. Indeed, with his heat (caloric) function (*a* − *a*′) discussed earlier as work, Carnot was also assuming the first law of conservation of energy as heat and work as well as the second law of spontaneous development of entropy. In fact, except for heat (*chaleur* or *calorique*), all these terms remained to be defined.

It is clear from his tentative equations that Carnot was already thinking in terms of the entropy–temperature product of negative Gibbs energy that is a key part of the viewpoint presented here. This idea was nascent in the concept of *calorique* he generally applied to the state of the working fluid and his pressure–volume form of the logarithmically variable *chaleur specifique* indicating the heat required. We conclude that this term should not be confused with current definitions of specific heat or heat capacity that are purposely restricted to the internal energy or enthalpy (*C*_v_ or *C*_p_). Furthermore, in Thurston’s 1890 (Macmillan) translation of Carnot’s book *Reflections on the Motive Power of Fire*, the translator (pointed out by Mendoza [2] Dover edition 1960) often used the same term heat for both of Carnot’s terms *calorique* and *chaleur*, providing lingering confusion regarding Carnot’s account. However, Carnot specifically states that while he is indifferent to the use of terms *chaleur* or *calorique* as a quantity of heat, he does reserve *chaleur* as a measure for the sensible heat of fire and is consistent in using *calorique* for changes in the state of the working fluid; we would consider the latter as variations in Gibbs energy or configurational entropic energy, not sensible heat.

At no stage is *calorique specifique* referred to in Carnot’s text despite a comment by Mendoza in 1960 [2] to the contrary, with *chaleur specifique* used consistently. It must be remembered that entropy includes an internal energy or enthalpy component not relevant to Carnot’s *calorique*. Carnot clearly distinguished this heat related to expansion and compression from the modern heat capacity where there are no such processes in measurement. So, he was also proposing the first law of the conservation of energy as well as the second law of increasing entropy as shown in Equation (16). Carnot even identified the internal energy or enthalpy as a separate entity of heat *U* in his equation given here as (38), which is discussed in detail on page 43 of his memoir [2].
*s* = *e* + *U* = *T*′ log(*v*) + *U*(38)
where *s* is “the quantity of heat (*chaleur*) necessary to change the air that we have employed from the volume 1 and from the temperature zero (i.e., Celsius) to the volume *v* and to the temperature *t*, the difference between *s* and *e* will be the quantity of heat required to bring the air to the volume 1 from zero to *t*. This quantity depends on *t* alone; we will call it *U*”. To obtain motive power, Carnot varies Equation (38) with temperature. Since he had already explained that the difference in heat capacity at constant pressure and that at constant volume was a constant independent of the working substance (i.e., in modern terms, *C*_p_ − *C*_v_ = *R*), it is reasonable to conclude that *U* represented the internal energy (*E*) or the enthalpy (*H* = *E* + *RT*), and thus, Equation (38) is closely analogous to Equations (16) and (17) that express the 2nd law of thermodynamics and statistical mechanics. Our modern version of this equation has the benefit of both the clarifying work of Clausius [15] on correctly establishing the fundamental principles of the mechanical theory of heat, the statistical mechanics of Willard Gibbs [8], and the quantum theory of Planck [14] and even Einstein soon after 1900.

Referring to the equation for entropic energy (*ST*) or total reversible heat requirement from Equation (11) above for a monatomic gas, we observe that for argon, Equation (39) applies at 1 atm external pressure.
*S_n_T_n_* = *RT_n_*ln[e^5/2^(@_t_/*ħ*)*^3^*]= 2.5*RT_n_ + RT_n_*ln[(@_t_/*ħ*)*^3^*] = 2.5*RT_n_* + 3*RT_n_*ln[(n_t_)](39)

Thus, expressing Equation (39) between two temperatures provides variations in Gibbs energy that allows the expression of motive power or rate of work. Thus, the similarity of Carnot’s Equation (38) and our equation for entropic energy (39) is evident.

In terms of the action method, the change in temperature within the logarithmic expression of (*kTI*)^1/2^ in adiabatic stages 2 and 4 exactly offsets the change in density and inertia (*I*_t_ = *mr*^2^), a function of volume, so the relative action *mrv* remains the same. In the adiabatic processes, the change in Gibbs or Helmholtz energies is a linear function of the change in temperature only, as is the internal energy. These changes in energy (*E* or *H*) in the adiabatic stages cannot result in net work, as stage 2=>3 is the reverse of stage 4=>1, ensuring that the working fluid returns to the same stage at the completion of each cycle. So, in Equation (38), taken with his conviction that as *chaleur specifique* or specific heat would also change with temperature, unlike the constant heat capacity now designated as *C*_v_, we recognize that Carnot proposed the first version of the second law of thermodynamics.

### 7.3. Heat Capacity versus Specific Heat

So, was Carnot correct after all in his definition of specific heat as varying with pressure? On page 38 of the Dover edition [2], Carnot performs a thought experiment confirming his viewpoint. He contends why a piston containing air first heated with *a* units to 100 degrees at constant volume (*V*_1_) and then, expanded with heat, added *b* units at constant temperature to a larger volume (*V*_2_) versus expanding first at 1 degree at constant temperature to the same volume *V*_2_ with *b*′ units and then heated to 100 degrees with *a*′ units must be equivalent. “As the final result of these two processes is the same, the quantities of heat employed for both should be equal:*a* + *b* = *a*′ + *b*′, whence *a*′ − *a* = *b* − *b*′(40)

*a*′ is the quantity of heat required to cause the gas to rise from 1° to 100° when it occupies the larger volume, and a is the quantity of heat required when it occupies the smaller volume. The density of the air is less in the first case than the second and according to the experiments of MM Delaroche and Bérard, its capacity for heat should be a little greater”.

Carnot clearly considered that the measure of caloric content was a property of state, which is consistent with the future concept of entropy as a measure of heat content for both enthalpy and configurational energy as negative Gibbs energy; this must be the same initially and finally, assuming conditions for the cylinder contents are the same. The change in entropy per mole of argon is given by the following equation, with heat capacity *C*_v_ for argon remaining the same throughout as 1.5*R*. The energy change for an enclosed cylinder is equal to *C*_v_δ*T* in both cases.
Δ*S* = *R* ln (*V*_2_/*V*_1_) + *R* ln(*T*_2_/*T*_1_)^3/2^ = *R* ln[(*V*_2_/*V*_1_)(*T*_2_/*T*_1_)^3/2^(41)

However, the quantity of caloric *a* required for heating at *V*_i_ includes an extra amount for the change of temperature from *T*_1_ to *T*_2_, indicating a change in state and entropy. By contrast, expanding with *b* units at the lower temperature of 1 degree Celsius involves less heat required, given that it equals *RT* ln(*V*_2_/*V*_1_).

Carnot [2] continues on page 36, “The quantity *a*′ being found to be greater than the quantity *a*, *b* should be greater than *b*′. Consequently, generalizing the proposition, we should say: The quantity of heat due to change of volume of a gas is greater as the temperature is higher”. He also concludes: “The fall of caloric produces more motive power at inferior than at superior temperatures”, foreshadowing the benefits of having a heat sink for a given range of temperatures as close to absolute zero as possible, which is a hint regarding the third law of thermodynamics regarding zero entropy at absolute zero of temperature. However, Carnot had not completed his analysis of the effects of temperature versus pressure or system volume, and the truth of his conclusions depends on whether ratios of temperatures and volumes do depend on their relative magnitude.

Modern textbooks assume that the factors *C*_v_ and *C*_p_ for heat capacity of gases are constants, although this is not true at either high or low temperatures because of quantum effects. If we consider 3.5*R* as the heat capacity of air *C*_p_ given its predominant composition of nitrogen, the difference in entropic energy as negative Gibbs energy −*G* per mole between 298.15 and 297.15 K at the standard pressure of 1 atmosphere can be calculated as follows.
δ*s* = *k*ln[(3*kT*_297.15_*I*_t_/3*kT*_298.15_*I*_t_)]^3/2^(2*kT*_298.15_*I*_r_/2*kT*_297.15_*I*_r_)
= *k*ln[(297.15/298.15)]^5/2^
= *k*ln(0.991636)
*s* = 3.5*k* − 0.008399*k*
*s* = 3.4916*k* per molecule = 29.029 J per mol per degree Kelvin

So, the variation in heat content as enthalpy shows very little change as a result of the increase in Gibbs energy (ca. 0.24%), which is so small it can almost be overlooked. If the pressure is increased to 64 atmospheres, the specific volume (*a*^3^) for an ideal gas at the same temperature will be 1/64 that at 1 atmosphere, so separation of the molecules will be one-quarter of that at one atmosphere, with a decrease in the translational moment of inertia to one-sixteenth. Will this change the Gibbs work factor and the heat capacity measured as the change in entropic energy between 298.15 and 297.15 K?
*s* = 3.5*k* + *k*ln[(3*kTI*_t_)^3/2^/*ħ*^3^][(2*kTI*_r_)/*ħ*^2^]

In fact, assuming ideal gas behavior, the change in entropy between these two temperatures will be exactly the same as before, despite the increased density, since the relative translational and rotational actions will be the same as before.
δ*s* = *k*ln[(3*kT*_297.15_*I*_t_/3*kT*_298.15_*I*_t_)]^3/2^(2*kT*_297.15_*I*_r_/2*kT*_297.15_*I*_r_)
= *k*ln[(297.15/298.15)]^5/2^= 0.0084*k*

Thus, the heat capacity is very little affected by marginal changes in pressure, irrespective of the density. With this understanding, Carnot’s inclusion of the heat exchanged isothermally in reversible work processes as his *chaleur specifique* or specific heat, now recognized as negative Gibbs potential, has validity. Note in these calculations that with pressure constant at one atmosphere, the temperature and specific volume vary inversely, or the product of temperature and number density (N) is constant. If we allow pressure to vary with the lower temperature, holding number density constant so that the moment of inertia of N_2_ is constant, heat capacity will vary proportional to kln(*T*_2_/*T*_1_)^5/2^ or 3.49160*k* per molecule or 29.02917 J per mole per degree Kelvin, which is the accepted value for *C*_p_ of N_2_.

By contrast, holding temperature constant, as in stage 1 of the Carnot cycle, the product of pressure and volume is constant (*PV* = *RT*), and the specific heat varies with the logarithm of the specific volume (or pressure), just as Carnot proposed in his table. This understanding of his term for specific heat is also relevant for the amount of heat consumed or released in stages 2 and 4 during adiabatic expansion and compression. As the entropic energy changes (Table 1, line 16; Table 2, line 31), some eight to ten times more specific heat is consumed or released to the fluid as Carnot’s caloric, depending on the logarithmic function of temperature and volume compared with the change in internal energy or enthalpy. So, we can conclude that in the same terms as Carnot hypothesized, the specific heat does vary with the volume, the temperature, or the pressure.

### 7.4. Quantum State Numbers

For both translational and rotational actions, mean values for quantum numbers (n_t_) supporting the molecular morphology of the field have been calculated in the tables, including Table 3, where these are summarized. These mean quantum numbers are calculated simply from the ratio of the action values (@_t_, @_r_) with Planck’s quantum of action (*ħ*). It is a feature of such translational quantum states that their magnitude decreases with the quantum number, as pointed out by Schrödinger [9], with only the levels very near the average energy occupied, explaining why the Maxwell–Boltzmann distribution has a sharp maximum. The average occupation number of the quantum cells in Table 1 and Table 2 approaches one in a million.

Note that none of the translational or rotational quantum states in heat engines correspond to extremely cold temperatures, so the quantum microstates are non-degenerate. This contrasts with the size of vibrational quanta that vary little with temperature, even down to absolute zero. Most texts in discussing quantum states are considering vibration, with the greatest population in the ground state, misleadingly for translational states. In Table 3, average quantum occupation numbers and magnitudes are shown. These are all exceeded by *kT* by at least an order of magnitude, even for rotation, indicating a lack of interaction between quantum particles as molecules of working fluid. As a symmetrical molecule lacking a dipole moment, nitrogen is proposed to exchange rotational quanta resonantly, with negligible net emission. The translational n_t_^3^ and rotational quantum j_r_^2^ products are also shown. For isentropic states, these are expected to be equal as adiabatic, although a small variation after four significant figures is shown in the table. Heat engines might function by irradiation with resonant quanta of specific long wavelength, varying according to the physical stage in the cycle, providing more efficient work than hitherto achieved. Such an experimental model should now be tested.

It is suggested that translational quanta are released gradually in molecular decelerations during elastic collisions. When recovering their velocities, the field quanta would be reabsorbed, only momentarily being freed. In effect, the Gibbs potential of separate molecules oscillates, being at a minimum nearest zero when colliding molecules are closest. In his discussion of Brownian motion, Einstein referred to forces acting between molecules in which this quantum field can participate. Action requires that all relative molecular motions be complemented by sustaining field energy related to configurational entropy. Yet the heat required for a molecular ensemble to reversibly reach a given temperature sustaining the enthalpy includes very significant heat contributions from changes in state such as melting or vaporization. This heat is regarded as consumed as work when the system reaches a given state under current environmental conditions of pressure and temperature. Whether such heat causing reversible changes of state disappears as work is performed or is consumed as field energy sustaining a molecular scaffold that also sustains external work (e.g., lifting a weight against gravity) is irrelevant; any such energy under given conditions can reappear as heat. This was Carnot’s conclusion.

The heat added or extracted in the isothermal stages in the cycle can be considered as changing the density of field quanta supporting the molecular field in expansion or compression. Since the internal and kinetic energy in these isothermal stages is constant, it has no overall role in the work performed. Action mechanics suggests that this irrelevance for internal energy may also be true of the two adiabatic stages, although the variation in the Gibbs energy in adiabatic stage 2=>3 (*b*′) of dilatation is greater than that in stage 4=>1 (*b*), the difference also being equal to the maximum work performed. This is a result of the higher quantum number for the former process and the reduced quantum number in stage 4=>1 after heat is lost isothermally in stage 3=>4. Obviously, the molecular kinetic energy dictates the momentum and pressure on the piston, but this would not be possible without the intensity of the quanta exerting the primary pressure on molecules with impulses at the speed of light. This is a basic statement of action resonance theory [4]. Figure 4 summarizes various variables and their changes in Carnot cycles. A flywheel is indicated that would be required to provide reversibility in the cycle. From maximum compression in stage 1, heat flows spontaneously from the hot source as marginal cooling occurs in expansion of the cylinder, as driven by the flywheel. Once the hot source is removed, adiabatic expansion doing external work continues, cooling the working fluid. The increased volume compensates for the decreased temperature with no change in action and entropy, but with a fall in Gibbs energy, which is shown as the red line in Figure 4. Reversing these stages by reversing the flywheel would absorb heat at a low temperature from body B, compressing the volume to reach the temperature of the hot source and slightly exceeding it. Then, heat is released to the hot source body A, acting as a heat pump from cold B to hot A.

If the working fluid expands irreversibly into a vacuum following heating by body A, as in a Joule or Gay-Lussac expansion, no work would be performed so no cooling need occur. That is, the cubic action (*mrv*)^3^ would increase proportional to the increase in volume (*r*^3^), but the temperature would remain constant, with no effect on mean molecular velocity (*v*). While the Gibbs energy would decrease by the increase in entropic energy on expansion, the action field would contain the same amount of field energy as before, since no heat is needed given the vacuum, but with a larger quantum number of smaller quanta n_t_ cubed. Thus, the size of the associated quanta must be diminished by the same amount as the radial separation is increased. The frequency of impulses would be decreased by the increased radial separation, but the torques developed would remain the same. Viewing kinetic energy as the statistical consequence of mean value of torques exerted in exchanges of quanta, the temperature will remain the same if well insulated from the environment at large.

This Joule expansion also has relevance for the famous Gibbs paradox, explaining realistically why combining two equal volumes of the same gas isothermally does not increase the total entropy, even if the two identical volumes of gas fully diffuse into one another by Brownian motion. Obviously, the mean negative Gibbs energy per molecule dependent on the mean action *mrv*, which was previously the same in both volumes of gas at the same pressure, will remain identical after the diffusion. This is an objective solution to the paradox that does not involve knowledge by an observer that all molecules are identical, assuming they are. However, if the two volumes of the same ideal gas differ isotopically in the number of nucleons, then both isotopes will have increased entropy on mixing and decreased Gibbs energy or chemical potential. The two isotopic species are considered to exist independently in their separate action fields. However, these differences are usually ignored in thermodynamics where calculations use averages.

## 8. Implications for Climate Science and Future Research

This action revision of the Carnot cycle restores the complementary relationship between kinetic molecular motion and field energy as a kind of quantum ether varying density. It is clear that the latent energy in a gaseous system associated with the Gibbs potential is quantum state energy, which is unavailable unless there is a change of state such as condensation or freezing, chemical reaction, or gravitational potential [12]. The greater the Gibbs potential, the lower the density of quantum field energy. The Gibbs potential must also be zero at zero Kelvin when all motions bar vibrations cease, but its value can only decrease while increasing negative values as the temperature increases and the molecular field gains quanta. The associated kinetic energy of molecules that also contributes to the total entropic energy is complementary to this quantum field but is not a contributor to the Gibbs potential. Whenever there is a change in the molecular configuration of the system such as in chemical reaction, there will be a reassignment of the quantum field so as to match the new configuration, with equilibrium occurring under conditions where Gibbs potential becomes equal in all phases. Thus, the melting and vaporization of a water system with phase equilibrium involves no change in Gibbs potential.

A critical process of the reversible heat engine cycles shown in both tables in stage 1 involves the absorption of radiant heat from the source by the molecular field accompanying external work, which is shown as a Gibbs energy decrease or an entropic energy increase. In the reverse cycle at stage 3=>4 at lower temperature during recompression, only part of this field energy stored in the field is returned as heat to the sink, increasing the Gibbs energy of the working fluid, the remainder being accumulated as reversible external work by each cycle. Carrying out external work by a heat engine in a gravitational field requires sustaining field energy, if the temperature is to be maintained. It is noteworthy from Figure 2 and Figure 3 that the decline in internal energy (*C*_v_δ*T*) is only a small fraction (ca. 10–15%) of the increase in Gibbs energy in the adiabatic expansions in Table 1 and Table 2. To ignore the transfer of such a substantial source of energy in the Carnot heat engine to external work is a significant omission. In fact, in Table 1 and Table 2, the density of translational molecular kinetic energy has been found to be very similar to the density of translational quantum energy, which is remarkably consistent as the molecular and quantum pressures (*kT*/*a*^3^ = *RT*/*V*_m_). Action resonance [4] proposes that there is just such an equation between the rate of change of momentum of molecules and the action force field of quantum exchanges. Demonstrated here as operative in the cylinders of the Carnot cycle, we can expect a similar relationship between molecular and quantum pressures in all physical environments. As explained earlier in the text, for monatomic gases and nitrogen (N_2_) discussed in this paper, vibrational entropy can be neglected, but in the general case, it will need to be considered for low frequencies. Furthermore, in the context of climate processes, we introduce another kind of entropy, the vortical entropy described below, to account for climate specific physical phenomena.

### Vortical Entropy

In our previous papers [7,12], we have used action thermodynamics to calculate the standard entropies of all atmospheric gases as a guide to better understand heat processes in climate. The virial theorem was invoked to provide a firmer basis for estimating the lapse rate with gravitational elevation, finding that 6.9 Celsius per km was the expected rate of change in temperature [12]. This value is very close to the observed lapse rate. We introduce the new concept of vortical entropy as a function of the air flow under Coriolis forces in anticyclones and cyclones, as illustrated in Figure 5 and shown in Table 4.

Vortical entropy is derived from vortical action in a similar manner as translational and rotational entropy. Vortical entropy is derived in Equations (42) and (43) for a parcel of air of N molecules of mass *m* rotating in an anticyclone or cyclone at *R* from the geographical center with angular velocity Ω (rad/sec).
@_vort_ = *mR*^2^Ω(42)
*S*_vort_ = N*k* ln(*mR*^2^Ω/*ħ*)(43)

This vortical action hypothesis proposes that a quantum energy field similar to that examined here for relative translational and rotational motion exists between the correlated groups of molecules operating as vortexes. In anticyclones, air circulation is almost linear near the extremities of the cells hundreds or thousands of km in magnitude, but it is inhibited by friction near the surfaces, reducing wind speed near the surface. We have proposed that the downwelling radiation required to balance the Kiehl–Trenberth model [18] is a function of surface friction, releasing radiative heat near the surface.

The action in Table 4 was calculated using mean values of properties for molecules in air (mass 29 daltons, 1 atmosphere pressure, and 288 K with bond length as the weighted mean for nitrogen and oxygen). Vortical entropic energy is effectively a higher scale of quantum field, which is justified by the action of the circulating air streams that is analogous to the translational relative action of molecules. Its estimated magnitude (Table 4) is greater than the sum of vibrational, rotational, and translational entropic energy, effectively raising the heat capacity of air in anticyclones. However, this thermal energy is released in turbulence at the Earth’s surface caused by friction. This solves the problem of how a cooler atmosphere can add heat to a warmer surface, seemingly defying the second law as defined by Clausius.

Environmental equity must also be considered. Substantial dissipation releasing radiant heat in temperate zones caused by such compressive obstructions may deprive environments at higher latitudes of beneficial warming from circulating air masses, even leading to incursions of polar vortexes. Given the significance of the Carnot cycle in defining thermodynamics, this revision of the hypothesis and the associated methodology for calculating the entropy and Gibbs energy of gases should have widespread application; this can help provide corrective solutions for the real risks such as those of climate change. These risks may require redefinition once the significance of heat reversibly stored in action fields such as *El Nino* is realized. We must be aware that action in ecosystems cannot be explained without understanding that the sensible heat of kinetic energy at all scales of molecular motion is sustained by complementary thermal radiation of much greater magnitude (*ST* versus *C*_v_*T*).

This hypothesis is amenable to experimental testing using radiometry, with sensors placed on agents disrupting laminar flows and generating turbulence and vorticity, such as large wind turbines and farms. A correlation with boundary layer vorticity and radiant heat release is predicted and possibly detectable from satellites. It is well known that the kinetic energy of molecules in laminar flow is not conserved at descending scales of turbulent motion. The larger heat content of atmospheric cells envisaged released in turbulence could explain the extremely hot conditions developed when large masses of air collide, which often precedes wildfires.

## 9. Conclusions

Despite the frequent omissions regarding Carnot’s exact contribution to general thermodynamics, the correctness of all his main postulates [1] have been vindicated in this action revision, as shown in the figures and tables.

“The maximum of motive power resulting from the employment of steam is also the maximum of motive power realizable by any means whatever… there should not occur any change of temperature which may not be due to a change of volume.”“The motive power of heat is independent of the agents employed to realize it; its quantity is fixed solely by the temperatures between which it is effected by the transfer of caloric.”“When a gas varies in volume without change in temperature, the quantity of heat absorbed or liberated is in arithmetical progression if the increments or decrements of volume are in geometrical progression.”“The temperature is higher during the movements of dilatation than during the movements of compression. During the former the elastic force of air is found to be greater and consequently the quantity of motive power during dilatation is more considerable than that consumed to produce movements of compression.”“The quantities of heat absorbed or set free in these different transformations are exactly compensated.”

Carnot’s main principle regarding the need irrespective of working fluid for it to vary in elastic force between a significant range of temperatures, his inference that the heat content (his specific heat) of the fluid would vary logarithmically with density, and his argument that all processes in the working fluid to maximize motive power would be reversible are all truly vindicated. Modern treatments of the Carnot cycle tend to emphasize the kinetic or internal energy and work performed as a function of differential variations with pressure as external work. As a result, they avoid mention of caloric, except to dismiss it as error. Carnot’s postulates once fully understood emphasize the role of that part of the entropy related to configurational or phase space and the statistical properties of entropy revealed by Gibbs [8] and Boltzmann [19]; irrespective of temperature or density, the entropy related to enthalpy governing internal energy retains the same value. Carnot’s conviction that that the specific heat of the air would nonetheless be a logarithmic function of volume, increasing with the radial separation of molecules as density declines and also of temperature, is vindicated in our action analysis. This is the basis of our proposal to include vortical motion of the atmosphere as an extended form for action, entropy, and extended latent heat content, suggesting a new area for research relevant to climate science.

Feynman [20] also attested strongly to Carnot’s accuracy in his Caltech lectures on physics in the early 1960s, even re-attributing the Clausius–Clapeyron equation relating vapor pressure and temperature as Carnot’s equation. We can now conclude it is the logarithmic variations in action and resultant Gibbs energy (or its inverse, entropic energy) between the two temperatures that control the motive power of the heat engine; the variation in enthalpy between the two temperatures is irrelevant to the motive power and work performed, although kinetic energy and pressure play a role. Obviously, for a reversible cycle, the Gibbs energies of the working fluid must always return to the same values in each stage, which is assisted by the inertia of the motor’s flywheel that ensures more heat is taken up at the higher temperature than is lost at the lower temperature. It should be noted that inertia in the engine also ensures that more caloric *a* is transferred from the hot source, preventing cooling during expansion, than loss of caloric *a*′ to the colder sink, with no change in entropy as the ratio of these values to the respective temperature.

It is anticipated that action revisions will contribute to better understanding of thermodynamics, allowing progress in non-equilibrium processes, the nature of irreversibility, and discussion of the maximum entropy production principle (MEPP). Although Boltzmann popularized the idea of entropy as disorder, it may be better understood as maximizing the freedom of action and the evolution of diversity. The energy field envisaged by Carnot in the concept of caloric can be regarded as a source of dynamic order for cyclic systems. Although the flow of energy in the solar system is an entropy-producing process overall, the surface of the Earth can be characterized more closely as in a steady state of entropy, oscillating locally in magnitude.

## Figures and Tables

**Figure 1 entropy-23-00860-f001:**
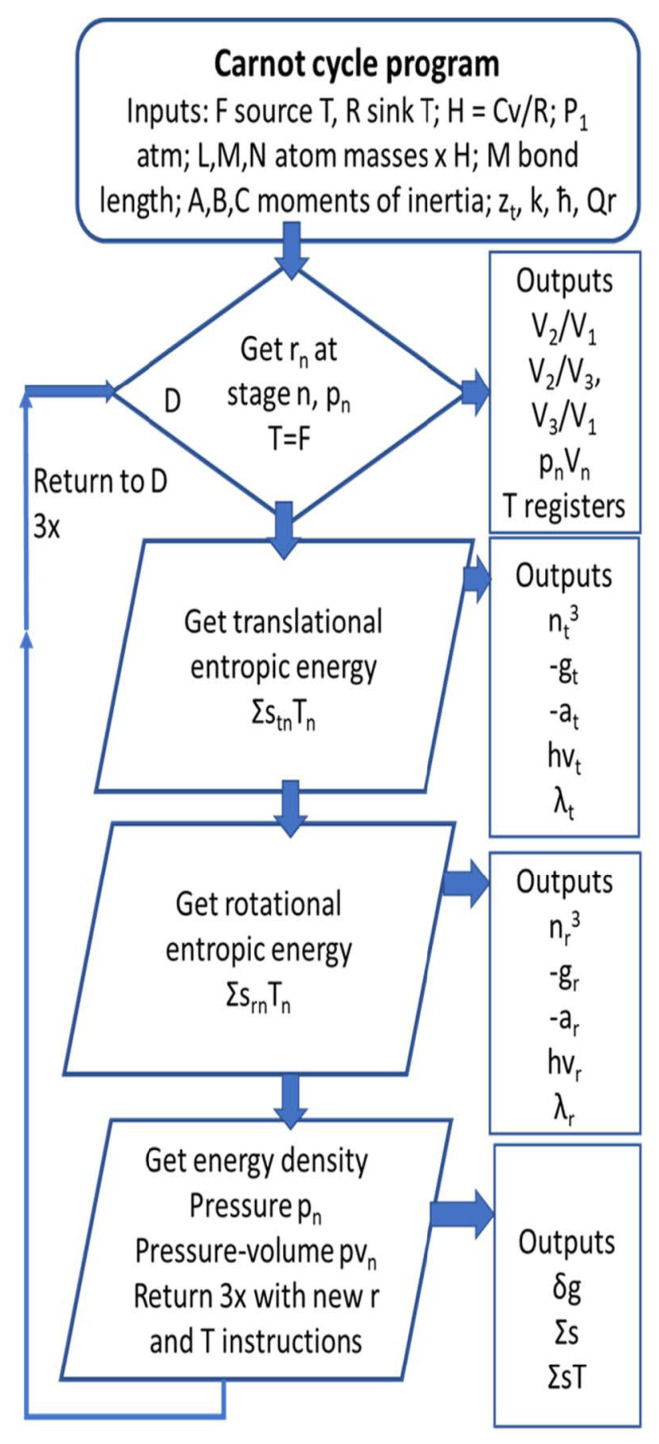
Carnot cycle flow sheet to estimate thermodynamic outputs. Run on Windows TRS32 emulator, Astrocal. See supplementary coding Carnot6/cal or contact the corresponding author.

**Figure 2 entropy-23-00860-f002:**
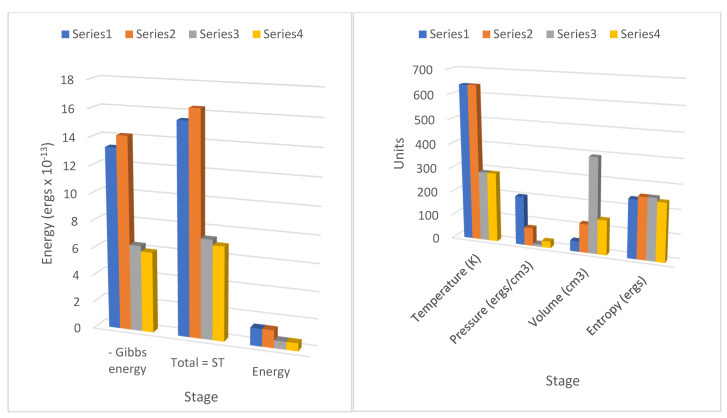
Action revision Carnot cycle for argon showing the thermodynamic energy in each stage commencing at 20 atmospheres pressure and 640 K temperature, with *kT* equal to Q_f_ initiating changes in action. All values for negative Gibbs energy (−**g*_t_*,) entropic energy (*sT*), and internal energy (*e*) are ×10^−13^ ergs per molecule. Heat Q_f_ is added from the heat source in the isothermal transition stage 1=>2 as shown in the increased values of *sT* and -*g*, and heat Q_r_ is removed to the sink in the isothermal stage 3=>4, showing markedly decreased value of *sT*, and −*g*. The value of the entropy per molecule is ×10^−17^ ergs/K, for temperature degrees Kelvin, pressure in Pascals, and volume per molecule ×10^−22^ cm^3^. Pressure, volume, and temperature conform to pa^3^ = *kT*. The small increase in entropy in stage 1=>2 of *k*ln(*V*_2_/*V*_1_) is equal to the decline in entropy in stage 3=>4, *k*ln(*V*_3_/*V*_4_).

**Figure 3 entropy-23-00860-f003:**
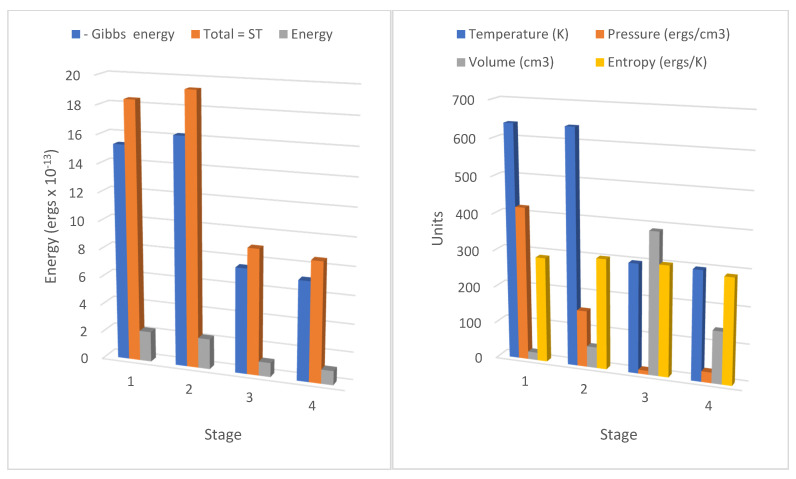
Action revision Carnot cycle for nitrogen showing thermodynamic energy in each stage; action is initiated in stage 1 at 40 atmospheres pressure and 640 K source temperature, declining to 298 K in stage 2. All values for negative Gibbs energy (−**g*_t_*,) entropic energy (*sT*), and internal energy (*e*) are ×10^−13^ ergs per molecule. The heat Q_f_ of *kT* is added from the heat source in the isothermal transition stage 1=>2 shown in the increased values of *sT* and −*g*, and heat Q_r_ is removed to the sink in the isothermal stage 3=>4, showing markedly decreased values of *sT*, and −*g*. The value of the entropy per molecule is ×10^−17^ ergs/K, for temperature degrees Kelvin, pressure in Pascals, and volume per molecule ×10^−22^ cm^3^. Pressure, volume, and temperature conform to *pa*^3^ = *kT*. The small increase in entropy in stage 1=>2 of *k*ln(*V*_2_/*V*_1_) is equal to the decline in entropy in stage 3=>4, *k*ln(*V*_3_/*V*_4_).

**Figure 4 entropy-23-00860-f004:**
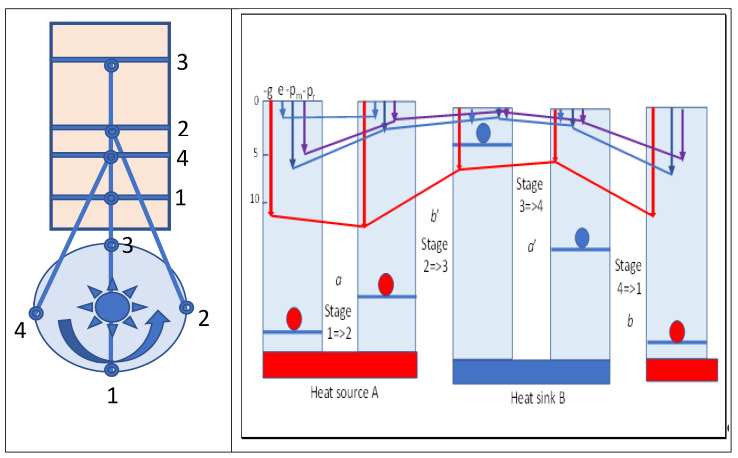
Carnot cycle for argon showing changes in mean molecular energy (*e*), Gibbs potential (*g*), pressure (*p*_m_), and quantum field intensity (*p*_g_). While the energy is stationary in stages 1=>2 and 3=>4, the Gibbs potential varies (*a*=>*b*′=>*a*′=>*b* in Carnot’s model), and thus, the maximum work possible is equal to (*a* − *a*′) = (*b* − *b*′).

**Figure 5 entropy-23-00860-f005:**
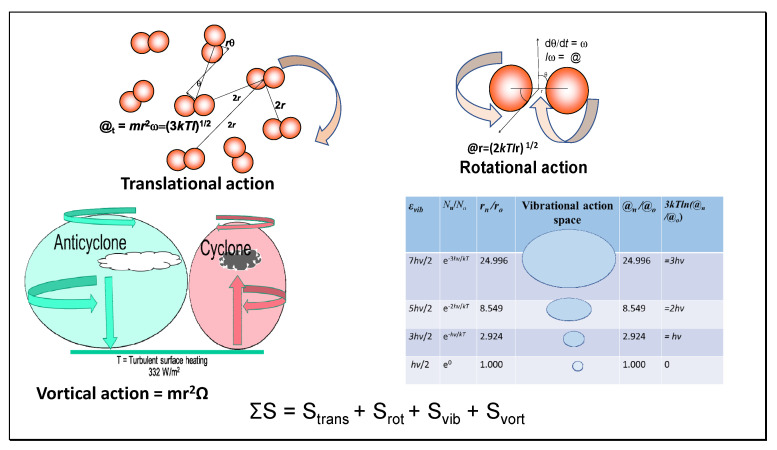
By analogy with translational, rotational, and vibrational relative action and entropy of N_2_ (see Equations (6), (7) and (9) respectively) shown here, we introduce vortical relative action in rotating air masses of anticyclones and cyclones, further increasing the action and entropy of masses of air and its heat capacity *ST*. Vortical action is modeled here simply as one-dimensional, but it would also contain a smaller fraction of second dimensional action but no third locally.

**Table 1 entropy-23-00860-t001:** Thermodynamic properties per molecule of monatomic argon in the cycle.

	Thermodynamic Property (Cgs Units per Molecule)	Stage 1=>2 Isothermal	Stage 2=>3 Adiabatic or Isentropic	Stage 3=>4 Isothermal	Stage 4=>1 Adiabatic or Isentropic
1	Degrees Kelvin	640 K	640 K=> 288 K	288 K	288 K => 640 K
3	Pressure (*kT*/a^3^) at t_o_	4.191891 × 10^7^	1.542111 × 107	2.094820 × 10^6^	5.694312 × 10^6^
4	Volume (*a*^3^) cm^3^ at t_o_	2.107880 × 10^−21^	5.729813 × 10^−21^	1.898111 × 10^−20^	6.9827762 × 10^−21^
4a	Relative volume	×2.718	×3.313	×1/2.718	×1/3.313
5	*pa*^3^ initial ergs	8.836006 × 10^−14^	8.836006 × 10^−14^	3.976203 × 10^−14^	3.976203 × 10^−14^
6	δ*pv* ergs	0	−4.8615 × 10^−14^	0	+4.8615 × 10^−14^
7	Translational inertia *mr*^2^ = *I*_t_ (g.cm^2^)	2.749749 × 10^−37^	5.355780 × 10^−37^	1.190173 × 10^−36^	6.110553 × 10^−37^
8	Action @_t_ (*Iω* erg.sec)	8.441202 × 10^−26^	1.178065 × 10^−25^	1.178065 × 10^−25^	8.441202 × 10^−25^
9	Quantum number	n = 80.042	n = 111.708	n = 111.708	n =80.042
9	*r*_t_ = *a*/2 (cm)	6.410895 × 10^−8^	8.947125 × 10^−8^	13.373586 × 10^−8^	9.556798 × 10^−8^
10	−*g*/*T* (erg/K)	1.815201 × 10^−15^	1.953263 × 10^−15^	1.953264 × 10^−15^	1.815201 × 10^−15^
11	−δ*g/T* ergs/K	+1.38062 × 10^−16^	0	−1.38062 × 10^−16^	0
12	−Gibbs energy (*-g*)	11.617287 × 10^−13^	12.500888 × 10^−13^	5.625399 × 10^−13^	5.227779 × 10^−13^
**13**	**δGibbs energy (δ*g*)**	**−0.8837 × 10^−13^ a**	**+6.875489 × 10^−13^*b*′**	**+0.39762 × 10^−13^*a*′**	**−6.389508 × 10^−13^ b**
14	Energy density (ergs/cm^3^)	5.5113594 × 10^8^	2.181727 × 10^8^	2.963682 × 10^7^	7.486693 × 10^7^
15	Total entropy	2.160358 × 10^−15^	2.298420 × 10^−15^	2.298420 × 10^−15^	2.160358 × 10^−15^
16	δ total entropy	+1.38062 × 10^−16^	0	−1.38062 × 10^−16^	0
17	Entropic energy ergs *s*_n_*T*_n_	13.826288 × 10^−13^	14.709889 × 10^−13^	6.619450 × 10^−13^	6.221830 × 10^−13^
18	δ*s*_n_*T*_n_	+0.8836 × 10^−13^	−8.090439 × 10^−13^	−0.3976 × 10^−13^	+7.604458 × 10^−13^
19	Net heat input	+8.839 × 10^−14^	+8.839 × 10^−14^	+4.8615 × 10^−14^	+4.8615 × 10^−14^
20	Internal energy *e*	1.325376 × 10^−13^	1.325376 × 10^−13^	0.59642 × 10^−13^	0.59642 × 10^−13^
21	Δ*e*	0	−7.28956 × 10^−14^	0	+7.28956 × 10^−14^
22	Heat transfer *Q_f_, Q_r_*	+0.8839 × 10^−13^	0	−0.3976 × 10^−13^	0
23	Entropy change=	+1.3812 × 10^−16^ = *Q_f_*/*T*_f_=3*k*ln(@_2_/@_1_)	0	− 1.3812 × 10^−16^ = *Q_r_*/*T*_r_ =3*k*ln(@_4_/@_3_)	0

**Table 2 entropy-23-00860-t002:** Thermodynamic properties of diatomic nitrogen Carnot cycle per molecule.

	Thermodynamic Property (Cgs Units)	Stage 1=>2 Isothermal	Stage 2=>3 Adiabatic	Stage 3=>4 Isothermal	Stage 4=>1 Adiabatic
1	Degrees K	640 K	640 K => 288K	288 K	288 K => 640 K
2	Pressure (*kT*/a^3^)	4.191891 × 10^7^	1.542111 × 10^7^	9.42669 × 10^5^	2.56244 × 10^6^
3	Volume (a^3^) (cm^3^)	2.107881 × 10^−21^	5.729813 × 10^−21^	4.21803 × 10^−20^	1.551725 × 10^−20^
4	Relative volume	×2.718	×7.362	×1/2.718	×1/7.362
5	*pa*^3^ (ergs)	8.836006 × 10^−14^	8.836006 × 10^−13^	3.976203 × 10^−14^	3.976203 × 10^−14^
6	δ*pv* (ergs)	0	**−0.48598 × 10^−13^**	0	**+0.48598 × 10^−13^**
8	*r*_t_ (cm)	6.410895 × 10^−8^	8.947125 × 10^−8^	1.740496 × 10^−7^	1.247120 × 10^−7^
9	Initial translational inertia *mr*^2^ = *I* (g·cm^2^)	1.924824 × 10^−37^	3.749046 × 10^−37^	1.4187304 × 10^−36^	7.2840050 × 10^−37^
10	Translational action @_t_ (*Iω* erg.sec)	7.062416 × 10^−26^	9.856396 × 10^−26^	1.118839 × 10^−25^	9.216142 × 10^−26^
11	Translational quanta	n_t_ = 66.968	n_t_ = 93.462	n_t_ = 121.963	n_t_ = 87.391
12	Translational –*g*_t_/*T*	1.741336 × 10^−15^	1.879398 × 10^−15^	1.989642 × 10^−15^	1.851580 × 10^−15^
13	−*g*_t_ (ergs)	1.114455 × 10^−12^	1.202815 × 10^−12^	5.730170 × 10^−13^	5.332550 × 10^−13^
14	Rotational inertia *I*_rn_	1.416704 × 10^−39^	1.416704 × 10^−39^	1.416704 × 10^−39^	1.416704 × 10^−39^
15	Rotational action @_r_	1.118839 × 10^−26^	1.118839 × 10^−26^	7.505400 × 10^−27^	7.50540 × 10^−27^
16	Rotational quanta	j_r_ = 10.609	j_r_ = 10.609	j_r_ = 7.117	j_r_ = 7.117
17	Rotational entropy	0.652131 × 10^−15^	0.652131 × 10^−15^	0.546600 × 10^−15^	0.546600 × 10^−15^
18	Rotational Gibbs energy	4.173639 × 10^−13^	4.173639 × 10^−13^	1.560635 × 10^−13^	1.560635 × 10^−13^
19	−(*g_t +_*g*_r_*_)_/*T* (erg/K)	2.393467 × 10^−15^	2.531530 × 10^−15^	2.531530 × 10^−15^	2.393467 × 10^−15^
20	−δ*g/T*	1.38063 × 10^−16^	0	−1.38063 × 10^−16^	0
21	−Gibbs energy ergs/molecule (−**g*_t_* −**g*_r_*)	1.531819 × 10^−12^	1.620179 × 10^−12^	7.290805 × 10^−13^	6.893185 × 10^−13^
22	**δGibbs energy**	**−0.88360 × 10^−13^ a**	**+8.910985 × 10^−13^*b*′**	**+0.39762 × 10^−13^*a*′**	**−8.425005 × 10^−13^ b**
24	Energy density	5.287088 × 10^8^	2.099222 × 10^8^	1.358496 × 10^7^	3.436531 × 10^7^
26	Energy density	1.979977 × 10^8^	7.283929 × 10^7^	4.205456 × 10^6^	1.143162 × 10^7^
27	Total entropy ergs/K	2.876686 × 10^−15^	3.014749 × 10^−15^	3.014749 × 10^−15^	2.876686 × 10^−15^
28	δ*s*_n_	+1.38063 × 10^−16^	0	−1.38063 × 10^−16^	0
29	Total entropic energy ergs/molecule *s*_n_*T*_n_	1.841079 × 10^−12^	1.929439 × 10^−12^	8.682476 × 10^−13^	8.284856 × 10^−13^
30	Energy density	7.267105 × 10^8^	2.827630 × 10^8^	1.728488 × 10^7^	4.442273 × 10^7^
31	δ*s*_n_*T*_n_	+0.8836 × 10^−13^	−1.031044 × 10^−12^	−0.39762 × 10^−13^	+0.983827 × 10^−12^
32	Net heat input	+8.836006 × 10^−14^	+8.836006 × 10^−14^	+4.721736 × 10^−14^	+4.721736 × 10^−14^
33	Internal energy *e*	2.208976 × 10^−13^	2.208976 × 10^−13^	1.028554 × 10^−13^	1.028554 × 10^−13^
34	Δ*e*	0	−1.180422 × 10^−13^	0	+1.180422 × 10^−13^
35	Heat transfer *Q_f_, Q_r_*	+8.836006 × 10^−14^	0	−3.9762 × 10^−14^	0
36	Entropy change=	+1.38063 × 10^−16^ = *Q_f_*/*T*_f_ = 3*k*ln(@_2_/@)_1_)	0	−1.38063 × 10^−16^ *= Q_r_*/*T*_r_ = 3*k*ln(@_4_/@)_3_)	0

For Table 1 and Table 2, it is assumed that each cycle operates from a starting cylinder pressure of 40 atmospheres at 640 K and is isothermally charged with extra heat *Q*_f_ = 1*kT* or 0.8836006 × 10^−13^ ergs per molecule during stage 1 when the piston is released. Then, −0.411427 × 10^−13^ ergs per molecule is transferred to the exterior refrigerator at 298 K during stage 3. Pressure is given from the perfect gas law, using *a*^3^ to indicate the cubic volume available to each molecule. The inertial radius *r* used in *I*_t_ is equal to *a*/2.

**Table 3 entropy-23-00860-t003:** Field quantum states.

	Stage 1	Stage 2	Stage 3	Stage 4
Argon, translational −*g*_t_, ergs	11.6173 × 10^−13^	12.5009 × 10^−13^	5.6255 × 10^−13^	5.2278 × 10^−13^
Mean quantum number	n = 80.042	n = 111.708	n = 111.708	n = 80.042
n_t_^3^	5.12807 × 10^5^	1.393968 × 10^6^	1.393968 × 10^6^	5.12807 × 10^5^
Mean quantum, ergs	2.2654 × 10^−18^	8.9679 × 10^−19^	4.0356 × 10^−19^	1.0194 × 10^−18^
N_2_, translational −*g*_t_ ergs	11.1446 × 10^−13^	12.0282 × 10^−13^	5.7302 × 10^−13^	5.3326 × 10^−13^
Mean quantum number	n_t_ = 66.968	n_t_ = 93.462	n_t_ = 120.584	n_t_ = 86.402
n_t_^3^	3.00346 × 10^5^	8.16404 × 10^5^	1.753352 × 10^6^	6.45017 × 10^5^
Mean quantum, ergs	3.7106 × 10^−18^	1.4733 × 10^−18^	3.2681 × 10^−19^	8.2674 × 10^−19^
N_2_, rotational −*g*_r_ ergs	4.1736 × 10^−13^	4.1736 × 10^−13^	1.5606 × 10^−13^	1.5606 × 10^−13^
Mean quantum number	j_r_ = 10.609	j_r_ = 10.609	j_r_ = 7.239	j_r_ = 7.239
j_r_^2^	112.551	112.551	52.403	52.403
Mean quantum, ergs	3.7082 × 10^−15^	3.7082 × 10^−15^	2.9781 × 10^−15^	2.9781 × 10^−15^
n_t_^3^ × j_r_^2^	3.3802660 × 10^7^	9.1887008 × 10^7^	9.1881105 × 10^7^	3.3800921 × 10^7^

**Table 4 entropy-23-00860-t004:** Entropy and negative Gibbs potential of air, including vortical potential energy.

288.2 K	Vibrational Action	Rotational Action	Translational Action	Vortical (ω = 5 × 10^−5^; r = 10^8^ cm	Vortical (ω = 5 × 10^−5^; r = 10^5^ cm	Vortical (ω = 5 × 10^−5^; r = 10^2^ cm
Action ratio (@/*ħ*)	<0.1	8.1	152.2	2.28259 × 10^15^	2.28259 × 10^9^	2.28259 × 10^3^
Entropy ln(@/*ħ*)	<0.01	4.18*k*	15.07*k*	35.364*k*	21.549*k*	7.333*k*
Energy	kJ per mol	10.017	36.115	84.749	51.642	18.532

## Data Availability

All data is contained within the article.

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
