# Peer review of "Action and Entropy in Heat Engines: An Action Revision of the Carnot Cycle"

_entropy, 2021, doi:10.3390/e23070860_

Round 1

Reviewer 1 Report

In this paper the authors proposed an "action" of each molecule of the working substance to describe the classical Carnot engine cycle. It is shown that the entropy per molecule can be written as a function of action. Therefore, the Helmholtz energy as well as Gibbs energy can be expressed by action. And the results of classical Carnot engine cycle can be well reconstructed. It is interesting to see that during the four thermodynamical process of Carnot cycle, the monatomic gas and diatomic gas shows quite different properties in terms of action. The main result is that the entropy of the working substance is related to the configurational or phase space of the thermostate, whick may be helpful to the relevant discussion about climate science. Those viewpoints are suitable for publication. However, in order to further understand the second law of thermodynamics by the concept of "action". I suggest the authors make a discussion about the irreversibility in terms of molecule action. 

Reviewer 2 Report

I clicked No Answer on the Overall Merit of the article for the following reason: While I agree with the thesis of the article that the standard take on Carnot's contribution is wrong and the article offers a new way to interpret Carnot's historic breakthrough away from the deterministic world of Newton to the changing-statistical world of Clausius and Boltzmann--I think that an average reader may not understand easily the authors' interpretation of Carnot Heat Engine. Part of the problems is that the article goes beyond a sharp focus of its thesis by including materials on action theory and its many implications, particularly Section 8.

It may be that the authors did not do a good job in explaining the action theory or, as I suspect to be the case, that the action theory--though the goal of the theory for correcting the deficiencies of current biological/ecological theories is admirable--is yet to be developed into a sound theory achieving its noble goal.

Specifically, I have the following suggestions: (1) deleting Section 8, which is not a necessary part of the article; (2) checking typo: on line 110, Lod should Lord.

I also have the following questions: In the book by one of the authors, Action in Ecosystems, you wrote, "Unlike Gaia, action resonance theory does not propose that any major conceptual change occurred when organic life appeared on earth, such as a reversal of the outcome of increasing disorder and entropy predicted by the second law of thermodynamics as time progresses." Since the main goal of action theory is to understand life, how does the theory treat the event of the appearance of life on the Earth. ALSO: would the authors care to comment on the MEPP (maximum entropy production principle) in the context of the action theory?

Reviewer 3 Report

The authors studied the Carnot engine cycle by using action theory. They derived the expressions of the work and heat along the isothermal and adiabatic branches. Derivation of  the work and heat in terms of mean values of action may provide an insight into understanding open systems as well as Carnot cycle,  though the Canot efficiency is independent of the working substance for the reversible Carnot cycle and the model under consideration is reversible.   However, the paper brings some confuse to me as a reader, and some issues should be clarified before I can recommend it for publication in Entropy.  
Some  issues need to be addressed:

(1) I suggest that the action theory should be briefly discussed/reviewed, instead of revisiting the Carnot cycle in the text book (see Sec. 2).

(2) The physical explanation of Eq. (13)and Eq. (32) should be given. The heat capacity is independent of system volume and temperature? Can one understand this with context of action theory?

(3) What is the physical meaning of “mean quantum state”or “quantum mean number”. Is it a concept that can be understood in statistical mechanics or in quantum mechanics?

(4) The paper introduces the concept of quantum. But, I can not find that the approach used here is within context of quantum thermodynamics. Could the authors make the difference between the classical approach and the quantum version here?

Reviewer 4 Report

In the present manuscript, a perspective on Carnot cycle is presented based in waht the authors call "action mechanics". As a physicist, I have a serious difficulty at understanding this work. First of all, I think that the text is filled in many places with unnecessary (and in many cases, questionable) jargon. Second, I see continuos self-reference regarding action mechanics. I didn't know about it in the beginning, but after revising some previous papers I have doubts about raising it to the level of a theory since it looks to me more, and I say it respectfully and acknowledging that I may be wrong, like a rewriting of known things with some incorporated numerology. A physical theory, we should agree, is much more than that. 

I don't think that the paper is clearly written. In several places, language is obscure and I don't get the point. Of course, I may be wrong in my appreciation (I'm being honest by saying that I don't understand the paper and it may be in part my fault) but in any case the manuscript is not easy to read, some equations come out of the blue, there are mathematical expressions that are not fully clear due to referencing problems, etc.

Below I summarize some passages where, I thing, the authors either get the physics wrong or say something difficult to understand. The list is not extensive, and there are many more issues, but these I'm citing are enough to make me refrain to suggesting publication of the present paper.

  1. Already in the abstract, line 17,  the authors say: "The Carnot principle shows that the maximum rate of work (puissance motrice) possible from the reversible cycle is controlled by the difference in temperature of the hot source and the cold sink, the colder the better." I actually don know what the authors mean by "rate of work" since in Carnot's cycle work is extracted quasistatically, i.e.,  Carnot's cycle gives the theoretical maximal efficiency, but at zero power.
  2. In line 144, it is not clarified what H stands for, but if it is enthalpy, then the expression is clearly wrong, since what should go there is internal energy. In any case the notation is not clarified by the context and is either confusing or wrong, as argued above.
  3. Minor detail: The sum sign in Eq. (10) seems to be misplaced.
  4. Line 203 "field energy", line 234 "mean virtual quantum size", line 235, "indicating the field energy as statistical work-heat"... Throughout the text, such esotheric denominations pop up without a clear sense of what it's being said. 
  5. I certainly don't understand the intermediate equality in Eq. (14). What does ν means in (14)? In particular, this intermediate equality in (14) implies (by separating nt) that something interpreted as a quantum number depends on temperature. I can't wrap my head around this unless these are averages of quantum occupation numbers, but the lack of statistical (and quantum mechanical) context in the paper is so strong that it is difficult to understand. 
  6. Line 246 : "these energies or functions for ideal gases have negative absolute values" I believe it is clear why this statement is conflictive. It is an issue of language: absolute values cannot be negative.
  7. Line 308: "The work done in an expansion at constant pressure is a logarithmic function of the volume ratio as shown in (21)" This an elementary (and serious) error. The work done in an expansion at constant pressure is trivially W=pΔV, the logarithmic dependence comes from constant temperature in an ideal gas.
  8. Lines 309-310: I don't understand why the authors claim that an expansion of the macroscopic volume at constant temperature implies a change in the configurational state of the molecules of the gas (more precisely, their moment of inertia). I would aaccept this somehow in the case of the (relatively) few molecules that are close to the expanding walls, but as interactions between walls and molecules, and between molecules themselves tend to be short ranged (otherwise thermodynamics should be seriously examined), I don't understand why configurational states of  molecules in the bulk should care about the expansion. This argument applies in many diffeent places in the text, I will just not mention it again, but it seems to be at the core of the authors' reasoning.
  9. Line 838 (and several other places in the text): "quantum number intensity" This shows that there is a real issue with language, which is why the work is, in my opinion, difficult to understand by a physicist. 
  10. Probably one of the most baffling propositions (from a physicist perspective) is that of a "quantum" vortical entropy to describe macroscopic (on scales from metters to kilometters) collective hydrodynamic flows. It is clear that at such energy scales, quantum effects are clearly not at all meant to manifest.
  11. I'm not sure to grasp the essence of the "international confusion" the authors claim to exist on Carnot's exact contribution to thermodynamics, nor how they have resolved it.

In summary, I don't feel moved to suggest publication of the present manuscript since there are many issues that, to my current understanding, don't fit together. Whether it is mere issue of language, or there are also conceptual issues, is not fully clear to me, but in the above discussion I have highlithed some conceptual problems that are serious enough as to proceed with caution.

Round 2

Reviewer 2 Report

The paper can be improved. However, the concept of action and its implication to "action thermodynamics" deserves to be heard by the science community.

Author Response

Thank you for your acceptance that the concept of action and its implications for thermodynamics deserves to be heard by the science community.

We note your opinion that the paper can be improved and will make some more modifications with this in mind. This is indicated in the letter to the editor.

Reviewer 3 Report

The authors have made  good reponses to my concerns in my previous report, and I can recommend the present paper for publication in Entropy now. 

Author Response

Thank you for your acceptance that we have met your concerns.

We want to acknowledge your help in improving the manuscript.

Reviewer 4 Report

I thank the authors for their detailed analysis of my main criticisms. Below, my take on what I consider the most important points in their responses (regarding my previous report). Let me first address some of the author's general comments before considering the direct answers to my criticisms.

  1. "Because of its derivation from a least action principle as expressed by Leibniz, Euler, Maupertuis and Gibbs, action providing a unifying principle across disciplines, including the translation between gravity and electromagnetism in physics, transition state kinetic theory, quantum and equilibrium states in chemistry and for information theory and molecular evolution in biology because of the possible structural role of configurational energy. The universal nature of this one action theory satisfies Occam’s razor, avoiding multiplicity in explanation. The Carnot cycle is just one primary example of how it can be applied."                                                                                                                                                                                                                                          I agree with the authors that action is a strong and helpful concept that pops out in many places in physics and that provides a natural and compact way to express laws of physics in terms of variational principles. I never denied that; what I do believe is that there is a lack of formality in the authors presentation that makes their discourse obscure. The claim that "this one action theory satisfies Occam's razor, avoiding multiplicity in explanation" is just a statement as far as it is not clearly backed up by what is written in the manuscript (and/or the responses). We should not take the author's word for it; it needs to be explicitly clear.  Sadly, I find this first answer particularly uninformative.                                                                                                            
  2. "The simplicity of estimating translational action, as precise values of mvr (with a symmetry factor of 2 to prevent double counting of couples), is conceptually appealing. This means the power of useful thermodynamics becomes available to a much larger audience and cohort, whereas it is actually falling into disuse, as calculus neglected by molecular genetics. The link between molecular action and thermodynamic entropy was well established in Kennedy et al. 2019 (Entropy 21,454-479). The fact that values of entropy for all atmospheric gases when calculated this way matched 3 rd law
    experimental values (i.e. the heat required to be added reversibly to achieve a particular T in Kelvin) to five significant figures is surely impressive."                                                                                                                                                                                                                                         
    I still cannot understand the relevance of what the authors define by "action" and this is the very least of my technical and conceptual concerns. The authors repeat several times in their responses and the manuscript that the action is mvr, but this is rather imprecise. As it seems, this is just the product of the mass of the type of particle in question (they are identical) by the mean speed and by the mean separation between particles. In the computation in the addendum to the authors response, they show how some thermodynamic potentials are naturally expressed in terms of what they call action (which is, in my opinion, a misleading name, in spite of this quantity having indeed dimension of action). But scrutinizing this computation it is not difficult to see that the authors are making a bold generalization from a very particular result (non-interacting quantum particles in a box). I invite the authors to try to perform the same computation with interacting particles and see if their action (as it is defined) comes out so cleanly and if it is even relevant. My comment about "numerology" in my first report had this deep meaning. What is a concept or a theory useful for if its structure needs to be modified as soon as one moves from the model one use to build it? Considering a polydisperse system of interacting particles makes all this to break down, while the statistical mechanics from where the "action" came in the first place is still valid. I thus believe that, as I said in my previous report, "action theory" has not earned such status and do not deserve to be elevated to that position.

In my last response above I  summarized my main concerns with "action theory". I will now proceed with the author's take on my direct criticisms.

  1. "The frequency v in hv has been removed; on reflection the previous version of (14) is considered a premature development and more likely to have a cubic solution, to be developed later. All mentions of related virtual quanta in the text and tables in the paper have been removed. In any case, these were unnecessary in the context of describing negative Gibbs energy in this paper."                                                                                                                                                                                                               I fully agree with the authors that this reference to "virtual quanta" was an abuse of language and certainly misleading. I am rather intrigued by the motivations the authors had to introduce that idea, but even beyond that, the meaning of the central equation they have now eliminated is still important, because it involved nt.By the way, why the need of changing notation? If nt is just the action normalized by Planck's constant, this is just unnecessary and confusing.                         
  2. "No wall effects are proposed. The molecules undergoing reversible expansion isothermally become more separated (i.e. increased a 3 or 8r 3 ), affecting the action (mrv) by increasing radial separation (r), as explained above. In the isothermal compression at low temperature,
    the action decreases, by reduced r with v constant). Yes, this thinking about configurational sate and field energy changing as temperature or volume changes is very much ”at the core of the reasoning in the paper”. We would like a re-examination of the paper, taking this into
    account, with the reviewer accepting how varying temperature and volume alter the action state implied in mrv by the cubic effects of volume and temperature mentioned."                                                                                                                                                                                       
    This is one of the most important concerns I have and, in my opinion, is a crucial point. I wonder whether the authors have considered the relative importance of this quantum contribution to entropy changes in a macroscopic system of identical (and non-interacting) particles. For such quantum effects to start to play a role, one needs the thermal wavelength to be at least of the same order of the inter-particle distance.  For instance, for one mole of  nitrogen molecules at 300 K, quantum effects appear at a volume of the order of 10-8 m3, which is so small that Van der Waals forces would make the ideal gas approximation to break down much before, since for nitrogen the Van der Waals constant b=0,387 m3.  As may be clearly seen, my concerns about this issue are totally justified and the authors response is, in my opinion, is not satisfactory.                                                                               
  3. "Although quantum effects are only considered significant in electromagnetism, this surely is a function of the relative size of the quanta involved. The problem of quantum gravity is magnified by the extremely small forces concerned, implying quanta of long wavelength.
    Action theory states that impetus of quanta or energy in the field is exchanged between molecules, keeping them independent, effectively balancing pressure exerted from impulses exchanged at the speed of light with the pressure exerted by the much slower molecules. We conclude that, as with the translation and rotation of atmospheric gases, such quantum exchanges should also apply with molecules in concerted streaming relatively across the vortices of anticyclonic and cyclonic motion, despite the immense scale for r in mvr. Thus the same principle of relativity is used for vortical action as for translational and rotational action. Furthermore, this extra heat capacity predicted is a testable hypothesis. However, we accept that introducing this concept here may also be premature. As a compromise we remove Figure 6 from the paper together with its supporting text. We prefer that the Editor decide if Section 8 in part or whole should be included."                                                                                                                                             
    In this response, the authors bring back the concept of "the field" that they decided to put away before for being conflictive. I see much verbose with little justification. Speed of light and quantum effects have nothing to do with macroscopic hydrodynamic flows.  The authors say  "we conclude that, as with the translation and rotation of atmospheric gases, such quantum exchanges should also apply with molecules in concerted streaming relatively across the vortices of anticyclonic and cyclonic motion, despite the immense scale for r in mvr", but I have no clue on how they arrive to such conclusion. I believe that the simple estimation I made in my previous point is illustrative enough as to clarify why such conclusion is clearly incorrect.

Sadly, I consider these criticisms to be strong enough to not recommend publication of the present manuscript.

Author Response

We thank Reviewer 4 who continues to provide useful feedback; we contend, however,  that the comments do not seriously diminish the validity  of the manuscript. 

  1. On further reflection, the reviewer's objection to the term "action theory" as lacking formality or being too general or obscure can be accepted. We have replaced this phrase throughout the paper with more specific  alternatives such as "action revision", "action mechanics" and "action resonance". We do welcome the reviewer's recognition that "action is a strong and helpful concept". Indeed, we can show it has valuable cross disciplinary relevance.  In the context of this paper, we claim action mechanics provides a more holistic concept for analysis, avoiding the more reductionist expressions of temperature, volume and pressure replaces by the simpler formula mvr. We suggest this will prove amenable for many descriptions of biological processes such as streaming of cytoplasm, driven by the rotation of microtubules.       
  2. We claim from our evidence (Entropy, 21, 454) that  action mechanics as advanced in this paper is dimensionally precise for gases under the temperature and pressure conditions described in the Carnot cycle, indicating minimal degeneracy. These environmental conditions are similar to those in most of the biosphere, thus requiring no need to modify the  model advanced for gases at normal pressures. This also explains the accuracy of our published results of entropy values. Further, we see no reason why our approach should necessarily break down in application to polydisperse systems such as liquids. For example,  the structure of liquid water as rolling clusters of about 10 hydrogen-bonded molecules lends itself to action-entropy calculations based on translation and rotation of these clusters, given Brownian motion of particles in water reflects gas-like conditions (Perrin, Nobel Laureate 1922). While the mass m of  polywater clusters will increase between 20 to 10 times, falling with temperature, the radial separation and volume per particle will diminish about the same order, yielding a similar value for the action ratio (mrv/h-). Despite the H-bonding interaction, degeneracy fails to increase and Gibbs energy and entropy can be calculated using action mechanics. We will deal with this elsewhere.       
  3.   We have reintroduced the concept of translational and rotational quanta corresponding to negative Gibbs energy  in Table 3. The mean size of rotational quanta for dinitrogen around 10-15 ergs as calculated in the four stages of the Carnot cycle, is reasonable, obtained by dividing -gby jr2.  This implies that the 3-dimensional version of translational quanta, dividing -gby nt3 , of the order of 10-18 ergs, is also of reasonable magnitude for field energy. We therefore speculate that the use of cubic nt  as a 3-dimensional  quantum level is also a logical expression, by analogy with rotational quanta, to be tested elsewhere.
  4. We do not suggest that action mechanics can replace statistical mechanics, given the latter's ability to handle states of high degeneracy with multiple occupation of  microstates. Rather, the simplicity of mvr that the reviewer surprisingly scorns seeks to make Bose-Einstein statistical approaches more accessible for the vast majority of environmental conditions  where degeneracy does not apply or is minor.  The relative action as mvr meets nearly all real world conditions, at least approximately. Note that we define action as mvrδΦ and relative action per radian as mvr
  5. The reviewer also underestimates its range of new applications. For example, the table in the addendum shows how activated vibrational states for Nand CO2 can also be expressed as low probability states of translational action, moving with high inertia and relative action on longer radii generating more sustained, impulses in collisions. Thus, an excited internal vibrational state in a molecule equilibrates with a low frequency Brownian translational state, compared to the more common ground state molecules. We propose this as a hypothesis showing the utility of action mechanics --  one that can now be tested.  
  6. The reviewer also denies the importance of emission and absorption of radiant quanta in macroscopic hydrodynamic fluids such as in cyclones. Yet it is well known that the radiant heat of condensation of water vapor drives cyclones, with the hydrodynamic kinetic energy involving only a low percentage of cyclone's power. Particularly in turbulent conditions, radiation is necessary for  energy balancing as streams become molecules We arrive at our hypothetical conclusion regarding vortical entropy by simple analogy with translational and rotational entropy. Furthermore, this hypothesis is testable using surface radiometry correlated with satellite observations.     
  7. While we are disappointed that Reviewer 4 is so inflexible regarding our proposals we appreciate very much that critique.